# SNX5 promotes antigen presentation in B cells by dual regulation of actin and lysosomal dynamics

Fernanda Cabrera-Reyes[1,2], Teemly Contreras-Palacios[1], Romina Ulloa[1], Jorge Jara-Wilde[4,5] , Mia Caballero[6], Clara Quiroga[3] , Carmen G Feijoo[7] , Jheimmy Díaz-Muñoz[1] , María-Isabel Yuseff[1]

**B cells rapidly adapt their endocytic pathway to promote the uptake and processing of extracellular antigens recognized through the B-cell receptor (BCR). The mechanisms coupling changes in endomembrane trafficking to the capacity of B cells to screen for antigens within lymphoid tissues remain unaddressed. We investigated the role of SNX5, a member of the sorting nexin family, which interacts with endocytic membranes to regulate vesicular trafficking and macropinocytosis. Our results show that in steady state, B cells form SNX5-rich protrusions at the plasma membrane, which dissipate upon interaction with soluble antigens, whereas B cells activated with immobilized antigens accumulate SNX5 at the immune synapse where it regulates actin-dependent spreading responses. B cells silenced for SNX5 exhibit enlarged lysosomes, which are not recruited to the synaptic membrane, decreasing their capacity to extract immobilized antigens. Overall, our findings reveal that SNX5 is critical for actin-dependent plasma membrane remodeling in B cells involved in antigen screening and immune synapse formation, as well as endolysosomal trafficking required to promote antigen extraction and presentation.**

## Introduction

B cells form membrane protrusions and ruffles to survey secondary lymphoid tissues in search of antigens, frequently tethered to the surface of presenting cells (1). Upon interaction with immobilized antigens, B cells rapidly form an immunological synapse (IS), which involves a spreading response, enhancing interactions between B-cell receptors (BCR) and antigens, which are subsequently internalized for processing and presentation to helper T cells, thereby enabling complete B-cell activation (2, 3, 4). The IS is considered a dynamic interface where B cells reposition their centrosome, in a Cdc42-dependent manner, guiding the local transport of MHC-II+ lysosomes, which, upon secretion, can aid antigen extraction from the synaptic membrane into endolysosomal compartments (5, 6). Polarization of the microtubule network is coupled to actin cytoskeleton remodeling, which involves the translocation of the hematopoietic lineage cell–specific protein 1 (HS1) from the centrosome to the synaptic membrane enabling the recruitment of the Arp2/3 complex to promote actin polymerization and membrane cell spreading (7).

B cells internalize antigens into hybrid compartments containing early and late endosomal markers, indicating tight coupling of endosomal trafficking for antigen internalization and processing (8 Preprint). How B cells manage to couple plasma membrane remodeling to antigen recognition and processing remains poorly studied. Sorting nexins (SNXs), such as SNX5, have been described to play a crucial role in membrane remodeling and endosomal trafficking in several cell types (9). In macrophages, SNX5 regulates micropinocytosis, where it is recruited to enriched extensions of actin during lamellipodium formation, in close proximity to Arp3, regulating actin polymerization and dorsal ruffling (10, 11). SNX5 harbors a PX domain for binding to phosphatidylinositol phosphates and a BAR domain that coordinates membrane deformation and interactions with other SNX to regulate trafficking of the EGF receptor (EGFR) from the plasma membrane to intravesicular compartments (12). In addition, SNX5 participates in the formation of the retromer complex, essential to retrieval of cargos from early endosomes to Golgi apparatus or to recycling endosomes, or from late endosomes to Golgi apparatus (13). Retromer complex, including vacuolar sorting proteins (VPS) and sorting nexins (SNXs), and several complexes such as WASH and branched actin generate membrane tubules for cargo protein transport (13). In B cells, impaired retromer function results in lysosome dysfunction, affecting antigen processing and B-cell activation leading to defective antibody production (14). However, despite this insight, the specific role of SNX5 in antigen uptake and processing by B cells has not been evaluated.

[1]Laboratory of Immune Cell Biology. Faculty of Biological Sciences, Pontificia Universidad Católica de Chile, Santiago, Chile   [2]Department of Gastroenterology, Faculty of Medicine, Pontificia Universidad Católica de Chile, Santiago, Chile   [3]Cardiovascular Diseases Division. Faculty of Medicine, Pontificia Universidad Católica de Chile, Santiago, Chile   [4]Laboratory for Scientific Image Analysis SCIAN-Lab, Integrative Biology Program, Institute of Biomedical Sciences ICBM, Faculty of Medicine, Universidad de Chile, Santiago, Chile   [5]Biomedical Neuroscience Institute BNI, Faculty of Medicine, Universidad de Chile, Santiago, Chile   [6]Laboratory of Neurobiology of the Audition, Faculty of Medicine, Universidad de Chile, Santiago, Chile   [7]Fish Immunology Laboratory, Faculty of Life Science, Andres Bello University, Santiago, Chile

Correspondence: jheimmariana@gmail.com; myuseff@uc.cl

Here, we studied the role of SNX5 in endolysosomal trafficking and actin-dependent functions in B cells. Our work reveals that SNX5 is critical to promote actin remodeling involved in membrane ruffle formation in resting B cells and spreading responses during activation with immobilized antigens. Silencing SNX5 severely compromises lysosomal integrity and function and impairs their recruitment to the IS, resulting in decreased antigen extraction by B cells. Together, these results reveal a dual role of SNX5 controlling both actin dynamics involved in capturing antigens and lysosomal integrity and function, required for antigen processing, two essential hallmarks of the B-cell immune synapse.

## Results

### SNX5 is required for membrane ruffle formation and is distributed in antigen trafficking compartments in B cells

To observe the formation of membrane ruffles in B cells at an ultrastructural level, we analyzed a mouse B-lymphoma cell line (IIA1.6) under steady-state conditions by transmission electron microscopy (TEM) (Fig 1A). We observed that B cells project protrusion-type membrane extensions and sought to evaluate the cellular mechanism involved. We focused on a member of the sorting nexin family, SNX5, which regulates membrane ruffle formation and trafficking of endolysosomal vesicles (15). For this purpose, we performed live-cell imaging of B cells expressing SNX5-GFP or GFP alone, as a control, under steady-state conditions followed by activation with soluble antigen (Ag) as an immunocomplex (see the Materials and Methods section). Our results show that B cells constitutively form membrane ruffles, where SNX5-GFP becomes localized (Fig 1B, Video 1). Importantly, these membrane protrusions were not positive for GFP alone (Fig 1B, bottom, Video 2), confirming that not all cytoplasmic proteins are recruited to these structures. After the addition of antigens, SNX5-rich protrusions became less extensive, suggesting that they might be involved in antigen scanning and recognition. To further study this process, we analyzed the localization of SNX5 in non-activated and activated B cells using immunofluorescence. For this purpose, B cells were incubated with soluble antigen at 4°C (time 0) and at 37°C for 10 min (early activation), fixed, and labeled for SNX5. The results show that at time 0 of activation, SNX5 is located at the edges of the cell ruffles, close to the antigen at the surface. After 10 min, membrane SNX5 protrusions disappear and clusters of antigens located closely to SNX5 become visible (Fig 1C), suggesting that the location of SNX5 is coupled to antigen stimulation. Next, to directly assess whether SNX5 is required for ruffle formation in B cells, SNX5 was silenced using two siRNAs (A and B), achieving a decrease of 30% and 65% in SNX5 expression levels, respectively (Fig S1A and B). Considering that enhanced silencing was achieved with siSNX5-B, this interfering RNA was chosen for functional experiments. Next, ruffle formation was analyzed in control and SNX5-silenced resting B cells using differential interference contrast (DIC) microscopy to visualize membrane protrusions (Fig 1D, Video 3). We observed that SNX5-silenced B cells exhibited reduced ruffle formation compared with control cells, as measured by an increase in circularity for

siSNX5-treated cells (Fig 1E) similar to previously described effects in SNX5-silenced macrophages (11). Also, we labeled filamentous actin (F-actin) in control and SNX5-silenced B cells, because ruffle formation is an actin-dependent process, and we found that actin is present in the protrusions formed in control cells, which were shorter in siSNX5 cells (Fig S1C). Because reduced dorsal ruffles are associated with decreased pinocytic uptake of dextran in macrophages (11), we sought to evaluate whether this functional effect also occurred in B cells. For this purpose, B cells silenced for SNX5 were pulsed with dextran for 2 h and chased for 4 h, and the accumulation of 10 kD dextran in lysosomes was quantified using fluorescence microscopy (Fig S1D). After a 4-h chase, siSNX5-B B cells accumulated less dextran in LAMP1$^+$ lysosomes compared with control cells (Fig S1E). Thus, SNX5 regulates membrane ruffle formation and pinocytosis in resting B cells, consistent with previous studies in other cell types (10, 11).

Given our previous results, we explored whether SNX5 played a role in the transition between the search and BCR-dependent capture of antigens (Ag) by B cells. To this end, we studied the localization of SNX5 in resting and activated B cells. Cells were activated with soluble antigen after 0, 10, and 60 min, fixed, and labeled using specific markers for different endocytic compartments (16), including Rab5 for early endosomes and LAMP1 for late endolysosomes. The results show that in resting cells, SNX5 exhibits a vesicular distribution toward the center and edges of the cell, including membrane ruffle projections (Fig 1F). After 10 and 60 min of activation (early and late activation, respectively), SNX5 was located with Rab5 (Fig S2A and B) and LAMP1$^+$ endolysosomal compartments (Figs 1F and S2C), in proximity to vesicles containing Ag. The above was evaluated using Manders' coefficient M1, which considers the overlap of SNX5 intensity over the label of Rab5-YFP or LAMP1. As activation progressed (late activation), Ag accumulated in LAMP1$^+$ endolysosomal compartments at the center of the cell in proximity to SNX5 (Fig 1F). To improve the visualization of these structural changes, immunofluorescence images of B cells stained for LAMP1, Ag, and SNX5 were processed in 3D projections (Fig 1G). We selected lysosomes containing antigens based on their signal intersections, which were used to differentiate them from those that did not, regardless of the amount of antigen or the extent of the fluorescence signal. Subsequently, the selected lysosomal Ag+ compartments were intersected again with the SNX5 label, to identify lysosomal compartments containing both SNX5 and antigens, rather than relying only on a triple colocalization quantification. From the 3D model intersection, we quantified lysosome compartments containing either antigen or SNX5 or both. Our results show that the percentage of LAMP1+Ag+ compartments increased, relative to LAMP1+ compartments, after later time points of activation, as previously described (16) (Fig 1H). The percentage of LAMP1+ SNX5+ compartments that contained Ag+ also increased during activation (79.03–97.21%), relative to LAMP1+ SNX5+ compartments (Fig 1H). These findings suggest that after activation, SNX5 redistributed to lysosome compartments that contain antigen.

Collectively, these observations reveal that SNX5 localizes and participates in the formation of ruffles before B cells encounter antigen and becomes located in lysosomal antigen trafficking compartments during B-cell activation.

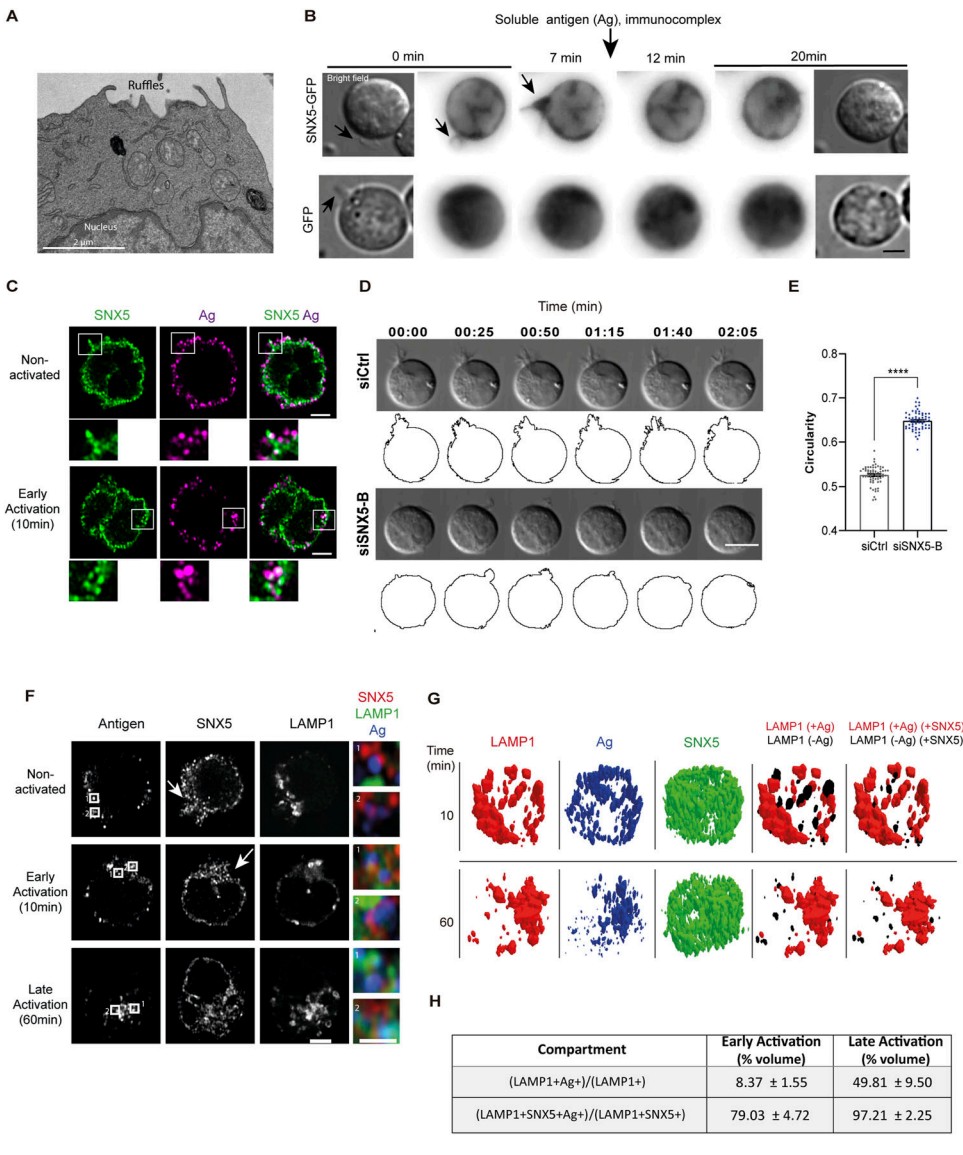

**Figure 1. SNX5 regulates membrane ruffle formation and pinocytosis in B cells.**
**(A)** Visualization of membrane ruffles in B cells under steady-state conditions by transmission electron microscopy. Scale bar: 2 $\mu$m. **(B)** Representative live-cell images via epifluorescence of B cells expressing SNX5-GFP and GFP as a control. Cells were incubated with soluble antigen after 10 min for different time points indicated. Images were acquired every 1 min and show a single plane. Arrows indicate ruffles. Bright-field images are shown at the beginning and end of the acquisition. Scale bar: 3 $\mu$m. **(C)** Representative confocal images of B cells activated with antigen (magenta) and labeled for SNX5 (green). Scale bar: 3 $\mu$m. **(D)** Representative images of Ctrl (siCtrl) or SNX5-silenced B cells using siRNA (siSNX5-B) acquired using differential interference contrast microscopy at various time points. Panels show segmented cells for each condition. Scale bar: 10 $\mu$m. **(C, E)** Quantification of the circularity index of images in (C) for two independent experiments, with n > 25 cells. A *t* test was conducted. **(F)** B cells were activated with antigen (blue) and labeled for endolysosomal compartments (LAMP1+, green) and SNX5 (red). A zoomed image of SNX5 (red), LAMP1 (green), and antigen (blue) from regions 1 and 2 delimited by a square is provided. **(G)** 3D volumetric model reconstructions for LAMP1 endolysosomal compartments containing uptaken antigen and SNX5 at early (10 min) and late time points (60 min). The image shows models for lysosomes (LAMP1, red), antigen (Ag, blue), and SNX5 (green) in activated B cells. The fourth panel shows lysosome models with antigen (LAMP1+Ag, red) and without antigen (LAMP1-Ag, black) volume, and the fifth (rightmost) panel shows LAMP1+Ag with SNX5 (red) and LAMP1-Ag with SNX5 (black). Representative models were selected from one cell. **(H)** Quantification of the volume percentage of Ag+ and LAMP1+ endolysosomal compartments (relative to the total LAMP1+ compartments) that contain antigens and also express SNX5. Center column: volume of LAMP1+ compartments containing antigen (LAMP1+Ag+) was selected and normalized by the total LAMP1 volume. Right column: volume of LAMP1+Ag compartments that intersected with SNX5 (LAMP1+Ag+SNX5), normalized by the volume of LAMP1 intersected with SNX5 (LAMP1+SNX5+). See the Materials and Methods section.

| Compartment | Early Activation (% volume) | Late Activation (% volume) |
|---|---|---|
| (LAMP1+Ag+)/(LAMP1+) | 8.37 ± 1.55 | 49.81 ± 9.50 |
| (LAMP1+SNX5+Ag+)/(LAMP1+SNX5+) | 79.03 ± 4.72 | 97.21 ± 2.25 |

## SNX5 localizes to the immune synapse in activated B cells

Having shown that SNX5 localizes to endosome and lysosome compartments, we next evaluated its role in protein trafficking during the activation of B cells with surface-tethered antigens. Under this condition, B cells form an IS and create a polarized contact site where antigen uptake and processing are tightly co-ordinated (6, 17). To determine whether SNX5 reaches the synapse, B cells were incubated with BCR ligand+ beads corresponding to F(ab')$_2$ goat anti-mouse IgG, for 0, 30, 60, and 120 min. BCR ligand− beads containing F(ab')$_2$ goat anti-mouse IgM were used as a negative control of activation. After each time point, the cells were fixed and labeled for SNX5 and $\alpha$-tubulin, to track the polarization of the MTOC to the IS together with the localization of SNX5 (18, 19,

20) (Fig 2A). After each time point of activation, the localization of SNX5 and $\alpha$-tubulin was quantified by defining a polarity index, which is a measure of the proximity of a fluorescent label to the bead using fluorescence microscopy. We observed that a subset of SNX5 was recruited to the IS concomitantly with centrosome ($\alpha$-tubulin) and accumulated at the antigen–bead contact site after 120 min of activation (Fig 2B and C), revealing that SNX5 becomes associated with the immune synapse during B-cell activation. To confirm this finding, B cells were activated with magnetic beads coated with a BCR ligand during the same time points of activation. The fractions of synapse-associated membranes were isolated using magnetic interactions with the beads, and their content was analyzed through immunoblotting (Fig 2D). Our results show that the accumulation of SNX5 in synaptic membrane fractions was

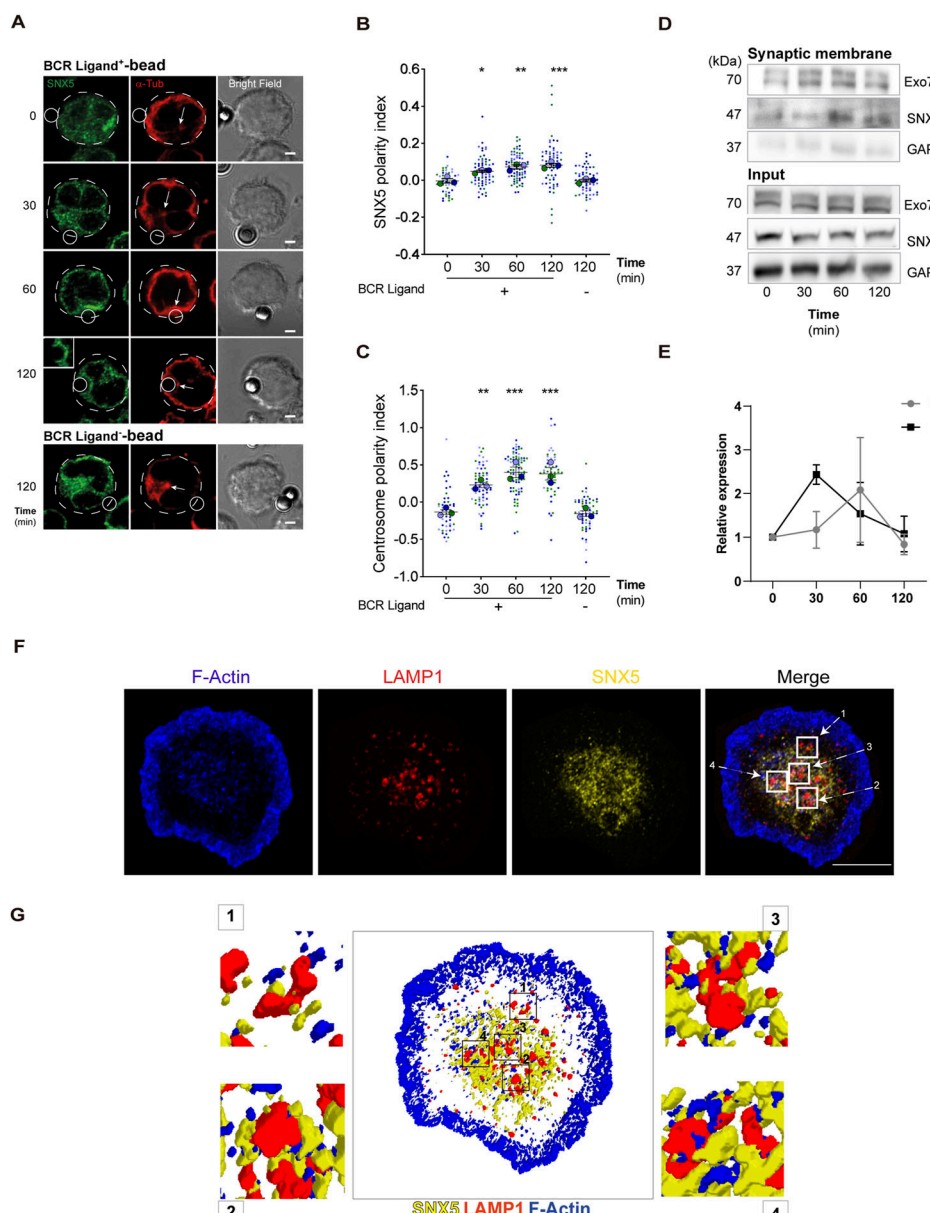

**Figure 2. B-cell receptor (BCR) activation triggers the recruitment of SNX5 to the IS.**
**(A)** Representative confocal images of B cells incubated with beads containing BCR ligand+ or BCR ligand⁻ (control beads) for different time points (min). Cells were fixed and labeled for SNX5 (green) and centrosome (α-tubulin, red). White circles and dashed lines indicate bead position and cell boundaries, respectively. White arrows indicate the position of the centrosome. The square on the left at 120 min shows a zoomed view of a bead. The scale bar in the bright field corresponds to 3 μm. **(B, C)** SNX5 and centrosome (α-tubulin) polarity indexes were calculated from the images in A at the IS for three independent experiments, n > 25 cells. 120 min (−) corresponds to cells incubated with control beads. One-way ANOVA followed by multiple comparison test under Tukey's criteria, * corresponds to the mean with respect to the control. *0.01 < P < 0.05; **0.001 < P < 0.01; ***P < 0.001. **(D)** Western blot analysis of synaptic membranes containing BCR-associated protein complexes isolated from B cells stimulated with magnetic beads for indicated times. Exo70, SNX5, and GAPDH were detected in samples. Representative images of two independent experiments. **(E)** Quantification of SNX5 and Exo70 normalized to input protein levels in total lysates. N = 2. **(F)** Representative confocal images of B cells incubated with antigen-coated coverslips for 60 min and stained for F-actin (blue), LAMP1⁺ compartments (red), and SNX5 (yellow). The stack where the immune synapse (IS) is located is shown (scale bars: 10 μm). **(G)** 3D reconstruction from immunofluorescence images shown in Fig 2F. A zoom image of SNX5 (yellow), LAMP1 (red), and F-actin (blue) from regions 1, 2, 3, and 4 delimited by a square is provided.
Source data are available for this figure.

detected after 60 and 120 min of activation, similar to Exo70 (Fig 2D and E), a component of the exocyst that had previously been shown to be recruited to the IS. Importantly, GAPDH, an unrelated cytoplasmic protein, was absent in this fraction (19). These results reveal that BCR activation promotes the accumulation of SNX5 at the synaptic membrane, and thus, we sought to characterize the localization of SNX5 in more detail within this domain. For this purpose, B cells were activated on antigen-coated coverslips for 60 min and stained for SNX5, LAMP1, and F-actin to analyze their distribution at the synaptic plane (Fig 2F). Immunofluorescence images were processed in 3D to visualize structures with better resolution (Fig 2G). We observed that SNX5 was found in close proximity to lysosomes and actin at the synaptic interface, suggesting that SNX5 could mediate interaction between both. We next

evaluated the localization of SNX5 with the BCR during the activation of B cells with surface-tethered antigens. To this end, B cells were activated on antigen-coated cover slides and the BCR was labeled after 60 min of activation (Fig S3A and B). Our results show that SNX5 associates with BCR clusters at the synaptic membrane (plane in contact with the antigen) and in the central plane of the cell, corresponding to an intracellular pool of the BCR (Fig S3A). Furthermore, we observed that SNX5 was associated with BCR-positive vesicles exhibiting tubulations (Fig S3B), a characteristic of antigen processing compartments previously described in dendritic cells (21). In addition, SNX5 partially localized with cortical microtubules at the IS and with microtubules under non-activating conditions, which may indicate a potential role in vesicle trafficking through this network (Fig S4A–C).

In general, these results show that SNX5 is recruited to the IS in close association with the trafficking of BCR–antigen complexes to LAMP1+ endolysosomal compartments suggesting that it could play a role in antigen extraction and processing during B-cell activation.

## SNX5 regulates the size of LAMP1+ endolysosomal compartments and their recruitment to the IS

Considering that our results show that SNX5 localized to LAMP1+ endolysosomal vesicles, we investigated whether SNX5 silencing affected the homeostasis of the lysosomal compartments, in terms of size and pH. For this purpose, LAMP1+ compartments in control and SNX5-silenced B cells under resting conditions were imaged using confocal microscopy and volumetric measurements of 3D reconstructions were obtained. The results reveal that the endo-lysosomal compartments in SNX5-deficient cells were ~50% larger in volume and exhibited overall sizes greater than 6 $\mu m^3$ compared with lysosomes from control cells (Fig 3A and B). The size of LAMP1+ compartments in B cells has not been reported before, but in macrophages that are larger cells, it ranges between 0.29 $\mu m^3$ and 5 $\mu m^3$ (22). Thus, silencing SNX5 in resting B cells appears to lead to an increase in the size of LAMP1+ compartments. In addition, we quantified the total number of LAMP1+ lysosomes per cell from 3D reconstruction, which does not change upon SNX5 silencing (Fig 3C), suggesting that an increase in lysosomal size is not due to an increase in their fusion.

Considering that SNX5 plays an important role in lysosome homeostasis, we evaluated the effect of SNX5 silencing on lysosomal pH, which was indirectly measured using LysoSensor. For this, control and SNX5-silenced B cells were incubated with LysoSensor Green DND-189, a probe that fluoresces at acidic pH, and live-cell images were acquired using epifluorescence microscopy (Fig 3D). A lysosomotropic agent, methyl ester of L-leucyl-L-leucine (LLOMe), which destabilizes lysosomes causing lysosomal damage and a decrease in pH, was used as a control to monitor a decrease in fluorescence intensity. The results show that the fluorescence intensity of lysosomes in SNX5-silenced B cells was similar to LLOMe-treated control-silenced cells, where both showed significantly less fluorescence intensity compared with the control cells. This result indicates that the absence of SNX5 leads to an increase in lysosomal pH (Fig 3D and E) and suggests that SNX5 regulates endolysosomal integrity in B cells.

As SNX5 regulates the size and pH of lysosomes under resting conditions, we evaluated whether their distribution to the IS was also impaired. For this purpose, SNX5-silenced B cells were activated with antigen-coated beads for different times and labeled for LAMP1 and γ-tubulin (Fig 3F). Next, the polarity indexes of lysosomes and centrosomes to IS were quantified in acquired images, revealing that silencing SNX5 impaired the recruitment of lysosomes to the IS of B cells (Fig 3G). In addition, cells did not exhibit the formation of a LAMP1+ ring around the beads (Fig 3H) (see the Materials and Methods section), where ring formation has been described to be associated with their docking and fusion with the synaptic membrane, which facilitates the extraction of antigens adhered to beads (6, 23). Of note, defective lysosome recruitment to the IS did not result from impaired centrosome polarization, which was not affected in SNX5-silenced B cells (Fig 3I). Thus, lower lysosome recruitment to the IS could be due to impaired association with microtubules. To further evaluate lysosomes at the synaptic interface and improve spatial resolution, B cells were activated on antigen-coated coverslips for 60 min (Fig 3J) and the number of lysosomes was quantified within the plane closest to the slide. Our results show that fewer lysosomes were recruited to the IS of SNX5-silenced B cells compared with their control counterparts (Fig 3K), consistent with decreased lysosome recruitment observed in B cells activated with beads (Fig 3F–H).

Overall, our results show that SNX5 is critical to maintain lysosome homeostasis in B cells. In addition, silencing SNX5 affects lysosome recruitment to the IS, suggesting that SNX5 could play an important role in immune synapse formation in B cells.

## SNX5 regulates actin-dependent B-cell spreading during the formation of an immune synapse

Considering that SNX5 was required for the formation of actin-based protrusions by B lymphocytes, we evaluated its role during B-cell immune synapse formation, where actin remodeling is critical to coordinate spreading responses (24) and cell polarity (7).

For this purpose, SNX5-silenced B cells were activated with antigen-coated beads for different times (Fig 4A) and F-actin around bead was quantified (Fig 4B). The results show that F-actin cup formation at the antigen–bead contact site was significantly decreased in SNX5-silenced B cells. Next, we evaluated actin organization and the capacity of control and SNX5-silenced B cells to spread progressively on antigen-coated coverslips for 30 and 60 min (Fig 4C). Our results show that SNX5-silenced cells exhibited decreased spreading responses compared with control cells, with areas decreasing by 13.2%, 23.1%, and 38.4% at 0, 30, and 60 min, respectively (Fig 4D). This finding is consistent with the role of SNX5 in actin remodeling in other immune cells, such as macrophages (11). In addition, B cells form actin foci at the IS, which are associated with BCR clusters to mediate internalization and extraction of antigens (25). We quantified the number of actin foci in SNX5-silenced B cells and observed a decrease in the number of actin foci at the synaptic plane (Fig 4E). We next evaluated whether silencing of SNX5 had an impact on BCR dynamics, considering that the formation of BCR clusters depends on actin remodeling (26). To this end, control and SNX5-silenced B cells were activated on antigen-coated coverslips for 10 and 30 min, fixed, and stained for F-actin and BCR (Fig 4F). BCR recruitment was quantified at the plane of the synaptic interface, where values close to 1 indicate recruitment toward the center of the synaptic plane and values close to −1 indicate a peripheral location. These results show that in control B cells, BCR recruitment to the center of the synapse increased over time (Fig 4F and G). However, in SNX5-silenced B cells, BCR recruitment to the center is delayed, showing mostly a peripheral distribution at both times of activation (Fig 4F and G). Altogether, our data suggest that the absence of SNX5 disrupts actin focus formation at the IS, which could be associated with lower BCR clusters, thereby affecting downstream signaling events essential for B-cell activation. Accordingly, upon activation with BCR ligand, SNX5-silenced B cells displayed lower levels of phosphorylated Erk

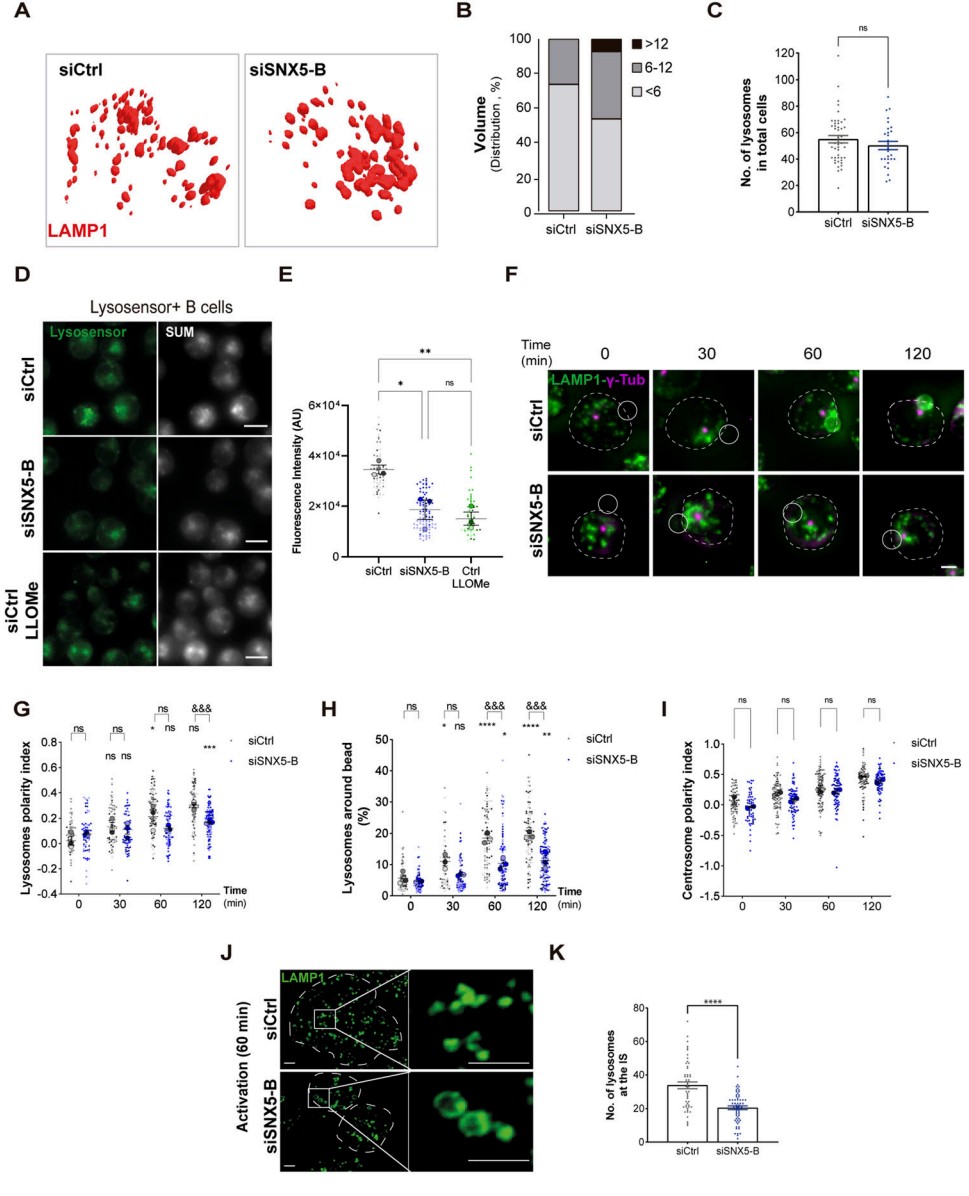

**Figure 3. Lysosomal integrity is compromised in SNX5-silenced B cells, triggering defects in their recruitment at the IS.**
**(A)** 3D volumetric model reconstructions for LAMP1+ endolysosomal compartments of Ctrl (siCtrl) or SNX5-silenced B cells (siSNX5-B). **(B)** Quantification of the volume of LAMP1$^+$ endolysosomal compartments per cell in 3D reconstructions from two independent experiments, n > 25 cells. Percentage distribution of lysosome volume per cell categorized as <6 $\mu m^3$, 6–12 $\mu m^3$, or >12 $\mu m^3$ is shown. **(C)** Number of LAMP1$^+$ endolysosomal compartments in Ctrl and SNX5-silenced B cells under steady-state conditions. **(D)** Immunofluorescence of Ctrl and SNX5-silenced B cells treated with LysoSensor Green. The positive control (Ctrl$^+$) refers to cells treated with 1 mM LLOMe for 60 min (scale bars: 10 $\mu m$). **(E)** Quantification of fluorescence intensity in z-projections of Fig 1D. One-way ANOVA followed by multiple comparison test under Tukey's criteria; * indicates the mean difference with respect to the control for three independent experiments, with n > 25 cells. Scale bars represent 10 $\mu m$. **(F)** Representative epifluorescence images of Ctrl and SNX5-silenced B cells incubated with antigen-coated beads for different time points, followed by staining for LAMP1$^+$ endolysosomal compartments (green) and centrosome (γ-tubulin). Images are presented as z-projections of a stack. White circles and dashed lines indicate bead position and cell boundaries, respectively. **(F, G)** LAMP1 polarity indexes were calculated from the images in (F) at the IS (see the Materials and Methods section) for the indicated times. Two-way ANOVA with Sidak's multiple comparison test was conducted for three independent experiments, with n > 25 cells. * corresponds to the mean with respect to its control. $^\&$ corresponds to the comparison of the mean between siCtrl and siSNX5-B. $^{\&\&\&\&}$ < 0.0001. **(H)** Quantification of the percentage of LAMP1$^+$ fluorescence around beads. Two-way ANOVA with Sidak's multiple comparison test was conducted for three independent experiments, with n > 25 cells. * corresponds to the mean with respect to its control. $^\&$ corresponds to the comparison of the mean between siCtrl and siSNX5-B. $^{\&\&\&\&}$ < 0.0001. **(F, I)** Centrosome polarity indexes calculated from the images in (F) for the indicated times. Two independent experiments, n > 30 cells. Two-way ANOVA with Sidak's multiple comparison test was conducted. **(J)** Representative confocal images of Ctrl and SNX5-silenced B cells incubated with antigen-coated coverslips for 60 min and stained for LAMP1 (green). Dashed lines indicate cell boundaries (scale bars: 3 $\mu m$). The stack where IS is located is shown. **(J, K)** Quantification of the number of LAMP1$^+$ compartments at the synaptic interface of images in (J) for three independent experiments, with n > 25 cells. A t test was conducted. * corresponds to the comparison of the mean between siCtrl and siSNX5-B.

(pErk) at later time points compared with control cells, suggesting that sustained BCR signaling could be regulated by SNX5 (Fig S5A and B).

We next sought to evaluate potential mechanisms behind the decrease in cell spreading and actin foci observed in SNX5-silenced B cells. SNX5 is recruited to enriched extensions of actin during lamellipodium formation, in close proximity to Arp3, regulating actin polymerization and dorsal ruffling in macrophages (10, 11). Thus, we focused on HS1, which was shown to become phosphorylated (pHS1) and recruited from the centrosome to the IS,

modulating spreading through actin polymerization by Arp2/3 (7). To evaluate whether the decrease in spreading observed SNX5-silenced B cells was due to a defect in the activation of HS1, we measured the levels of phosphorylated HS1 (pHS1) in control and silenced cells. For this purpose, B cells were activated on antigen-coated coverslips for 30 and 60 min, fixed, and stained for pHS1 (Fig 4H). Indeed, SNX5-silenced B cells showed lower levels of HS1 phosphorylation at the IS (Fig 4I), suggesting that SNX5 could regulate B-cell spreading by modulating HS1 activation during activation.

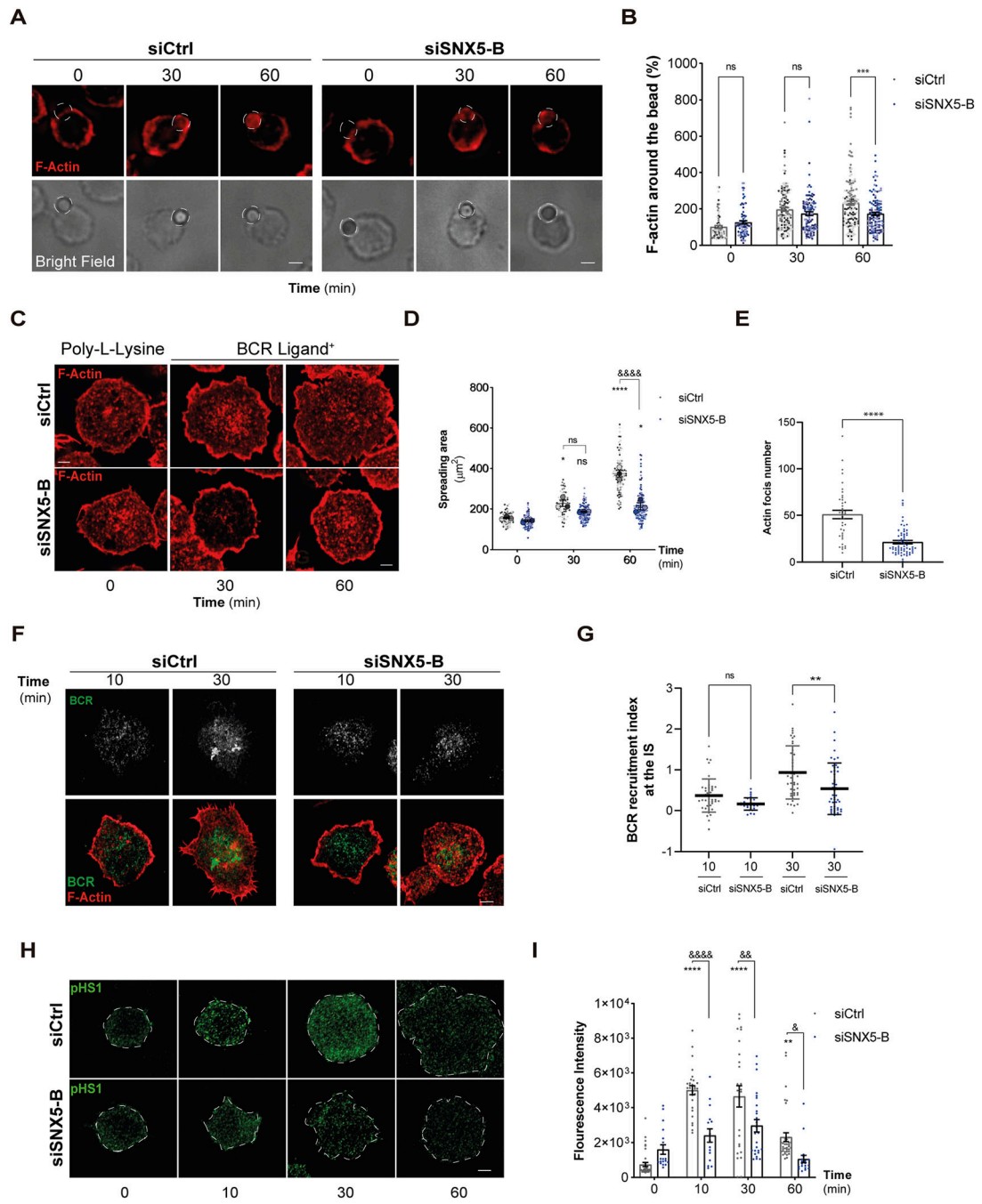

**Figure 4. Actin-dependent B-cell spreading during immune synapse formation is regulated by SNX5.**
**(A)** Representative epifluorescence images of Ctrl and SNX5-silenced B cells incubated with antigen-coated beads for different time points, stained for F-actin (red). Images are presented as z-projections of a stack. White circles indicate bead positions. The scale bar in bright field corresponds to 3 $\mu$m. **(B)** Quantification of the percentage of F-actin fluorescence around the bead. Values were normalized by the initial fluorescence (0 min). Two-way ANOVA with Sidak's multiple comparison test was conducted for three independent experiments, with n > 25 cells. **(C)** Representative confocal images of Ctrl and SNX5-silenced B cells seeded onto antigen-coated coverslips for 30 and 60 min and stained for F-actin (red). Non-activated cells were seeded on poly-L-lysine. **(D)** Quantification of the spreading area of F-actin ($\mu$m$^2$) from three independent experiments, with n > 30 cells. Two-way ANOVA with Sidak's multiple comparison test was performed. * corresponds to the mean with respect to its control. & is the comparison of the mean between siCtrl and siSNX5-B. &&&& < 0.0001. **(C, E)** Quantification of actin foci in B cells from the experiment shown in (C) at 60 min of activation, for two independent experiments, with n > 25 cells. A t test was conducted. **(F)** Representative epifluorescence images of Ctrl and SNX5-silenced B cells incubated with antigen-coated beads for 10 and 30 min stained for F-actin (red) and B-cell receptor (green). The stack where the IS is located is shown. **(G)** Quantification of the B-cell receptor recruitment index to the center of the IS. Two-way ANOVA with Sidak's multiple comparison test was conducted for two independent experiments, with n > 25 cells. **(H)** Representative confocal images of Ctrl and SNX5-silenced B cells seeded onto antigen-coated coverslips for 60 min and stained for p-HS1. The stack where the IS is located is shown. Dashed lines indicate cell boundaries (scale bars: 3 $\mu$m). **(H, I)** Quantification of fluorescence intensity of p-HS1 from images in (H). Two-way ANOVA with multiple comparison test under Tukey's criteria was performed for two independent experiments. * corresponds to the mean with respect to its control. & is the comparison of the mean between siCtrl and siSNX5-B. &&&& < 0.0001.

Overall, our findings reveal that SNX5 regulates B-cell spreading by activating the actin regulatory proteins HS1. In addition, SNX5 is required to promote the formation of actin foci, which could affect the correct positioning of LAMP1+ lysosomes and the subsequent extraction of antigens.

### SNX5 controls lysosome-dependent antigen extraction and presentation at the immune synapse

So far, we have shown that SNX5 plays an important role in actin remodeling at the immune synapse and that silencing of SNX5 results in decreased recruitment of lysosomes at the IS formed by B cells. In this regard, we next evaluated whether SNX5 also played a role in the association of LAMP1+ vesicles to actin. To this end, control and SNX5-silenced B cells were activated on antigen-coated coverslips for 60 min and the intersection between F-actin and LAMP1+ endolysosomal compartments was quantified in 3D processed images (Fig 5A). Our results show that the intersection between F-actin and LAMP1+ at the synaptic membrane decreased in cells silenced for SNX5 compared with control cells (Fig 5B). Considering that these cells also display fewer actin foci, we sought to evaluate the effect of SNX5 silencing on the extraction of immobilized antigen. To this end, control and SNX5-silenced B cells were incubated with BCR+ ligand–coated beads coupled to OVA for different time points (Fig 5C). Antigen extraction was evaluated by monitoring the remaining OVA fluorescence signal (percentage) on the beads associated with B cells (Fig 5C). The OVA signal decreases over time due to the progressive degradation of antigens facilitated by lysosome secretion (6). The results show that SNX5-silenced B cells were associated with beads exhibiting higher levels of OVA fluorescence compared with control cells, after 1–2 h of activation (Fig 5C and D), indicating that antigen extraction was compromised. We next investigated whether the reduced ability of SNX5-silenced B cells to extract immobilized antigens efficiently also affected their capacity to present these antigens. To do this, we incubated both control and SNX5-silenced B cells with beads conjugated to a specific BCR ligand with the Lack antigen from *Leishmania major*. We then assessed the ability of these B cells to present MHC-II–peptide complexes derived from the bead-bound Lack antigen to a specific T-cell hybridoma by measuring IL-2 secretion from activated T lymphocytes. SNX5-silenced B cells exhibited a significant reduction in the presentation of Lack antigen to T cells (Fig 5E). Notably, the presentation of the Lack peptide itself was unaffected (Fig 5F), indicating that SNX5 silencing does not impair B–T-cell interactions.

Next, we also evaluated whether the defects in antigen extraction and presentation in SNX5-silenced B cells were due to a failure in BCR internalization and turnover at the plasma membrane. To assess this, B cells were activated with Ag for 0, 5, and 10 min, where surface BCR was labeled under non-permeabilizing conditions (Fig S5C) to quantify the levels of BCR present on the cell surface. Flow cytometry analysis revealed that at steady state, the levels of BCR were similar in both control and SNX5-silenced B cells, suggesting that SNX5 does not participate in receptor turnover. It is worth noting that under resting conditions, the silencing of SNX5 did not produce changes in the total levels of BCR at the cell surface (Fig S5D). Therefore, these results suggest that defects in antigen

extraction were not due to a diminished ability of B cells to interact with the antigen but rather due to deficient lysosome recruitment at the IS.

## Discussion

The precise mechanisms by which B cells coordinate antigen uptake with processing in endolysosomal compartments are not yet fully understood. We here reveal that SNX5 plays a critical role in this process, by regulating endolysosomal homeostasis and actin-dependent spreading that affect antigen acquisition and extraction. Collectively, our results show that in resting cells, SNX5 is a protein that localizes within the endosomal system, where SNX5 partially localizes with the microtubule network, recycling, and early endosomes, which has been reported before in other systems (27, 28). SNX5 dimerizes with SNX6, which binds to the carboxyl-terminal region of the attached p150 subunit of dynactin (27), mediating vesicular transport through its binding to the motor protein dynein (29). This suggests that the SNX5 assembly could also be hetero-dimerizing with proteins that allow it to dock indirectly to the microtubule network to mediate vesicular transport.

When B cells survey the environment for antigens, they generate plasma membrane protrusions, where SNX5 is accumulated, similar to what occurs in macropinocytosis in macrophages (11). During activation with soluble antigen, SNX5 localizes to LAMP1+ endolysosomal compartments that contain internalized antigens. Thus, our results suggest that SNX5 moves between plasma membrane domains to promote protrusions and endolysosomal compartments to regulate antigen extraction and processing. Consequently, when SNX5 is absent, two critical processes are affected (Fig 6):

(1) Disruption of lysosomal homeostasis in B cells, which may be due to defects in the recycling of lysosomal components that occur on the surface of autolysosomes, where SNX5 was shown to be required to maintain lysosome regeneration (30). In this regard, we also propose that the absence of SNX5 may decrease tubulation in the autolysosomal membrane, hindering efficient lysosome formation and recovery, leading to the accumulation of autolysosomal structures. Interestingly, SNX8 has been reported to facilitate lysosomal tubulation during reformation, and its absence results in an increased lysosomal volume (31), similar to the phenotype we observed here. Thus, the participation of other SNXs in our model cannot be excluded. In addition, the absence of SNX5 results in functional changes in the lysosome, which were less acidic and were unable to be recruited efficiently to the IS compromising the capacity of B cells to extract and process antigens. Interestingly, it has been described that SNX5 interacts with VPS13A to mediate its association with endosomes and coordinate the trafficking of lytic granules in HeLa cells (32), where the absence of VPS13A also produced defects in endocytic trafficking and lysosomal degradation of cargos contained in autophagic and endocytic vesicles (33). It is most likely that the absence of SNX5 in these cells also reduces contacts of VPS13A with endosomes affecting the trafficking of enzymes required for antigen processing and lysosome function. However, the role of SNX5-VPS13A remains to be explored in B cells. Furthermore, the reduced recruitment of LAMP1+

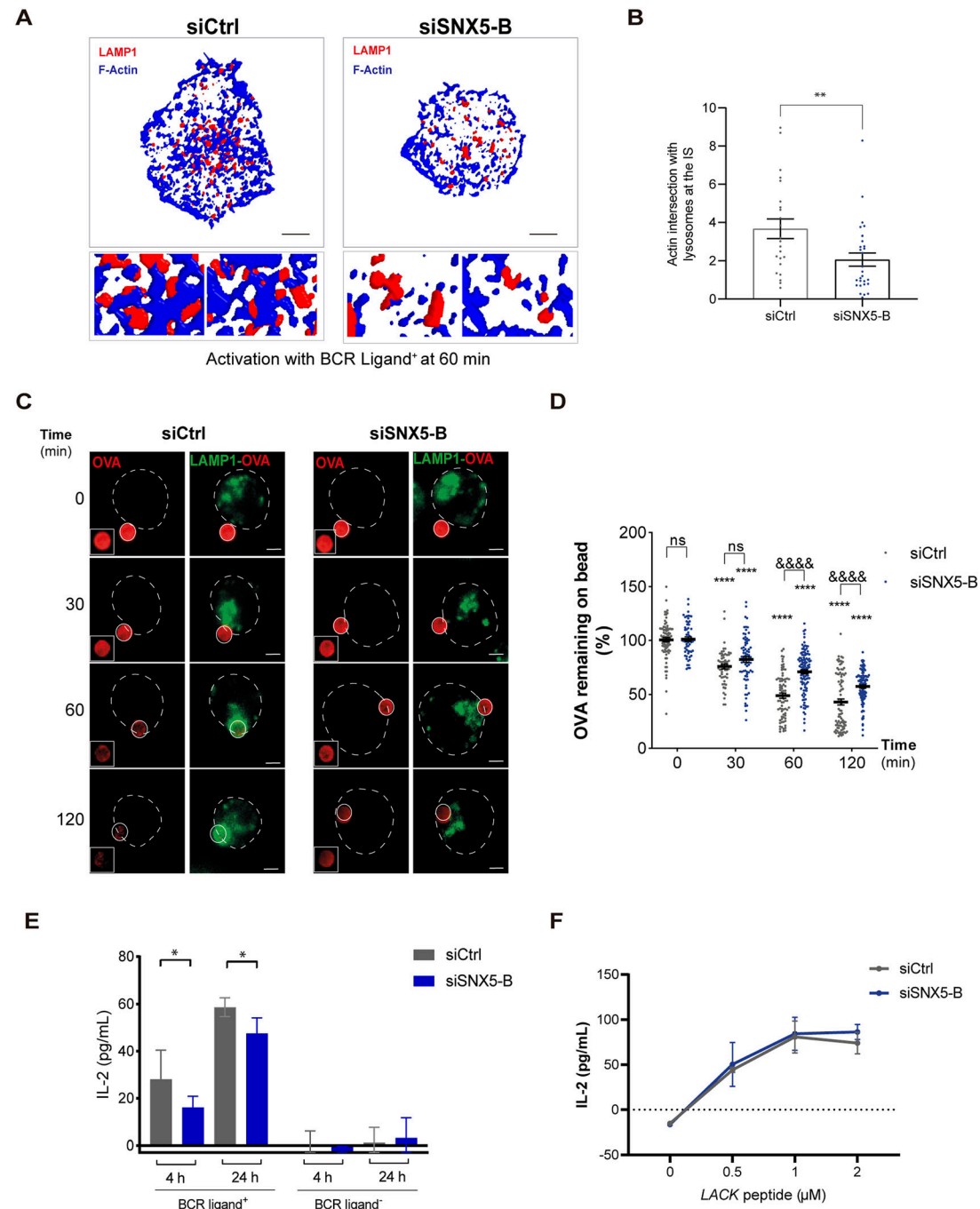

**Figure 5. Impaired antigen extraction in SNX5-silenced B cells.**
**(A)** 3D volumetric model reconstructions for LAMP1 endolysosomal compartments (red) and F-actin (blue) of Ctrl or SNX5-silenced B cells under activation with B-cell receptor ligand⁺ at 60 min. The stack where the IS is located is shown. **(B)** Quantification of actin intersection with LAMP1⁺ endolysosomal compartments at the IS, n > 23 cells. A *t* non-parametric test (Mann–Whitney U) was conducted. **(C)** Representative epifluorescence images of Ctrl and SNX5-silenced B cells incubated with antigen-coated beads coupled to OVA protein for different time points. Cells were stained for OVA (red) and LAMP1 (green). Images are presented as z-projections of a stack. White circles indicate bead positions and cell boundaries, respectively. **(D)** Quantification of the percentage of OVA fluorescence remaining on beads and lysosomes around beads, after incubation with B cells during different time points. Values were normalized by fluorescence at time 0. Two-way ANOVA with Sidak's multiple comparison test was conducted for three independent experiments, with n > 25 cells. * corresponds to the mean with respect to its control. & corresponds to the comparison of the mean between siCtrl and siSNX5-B. &&&& < 0.0001 (scale bars: 3 μm). **(E, F)** Antigen presentation assays. **(E)** Antigen presentation assay with control and SNX5-silenced cells. **(F)** Peptide control for the cells used. Mean amounts of IL-2 are shown for a representative image of two independent experiments performed in triplicate for control and SNX5-silenced cells. Two-way ANOVA with Sidak's multiple comparison test was conducted. * corresponds to the mean with respect to its control.

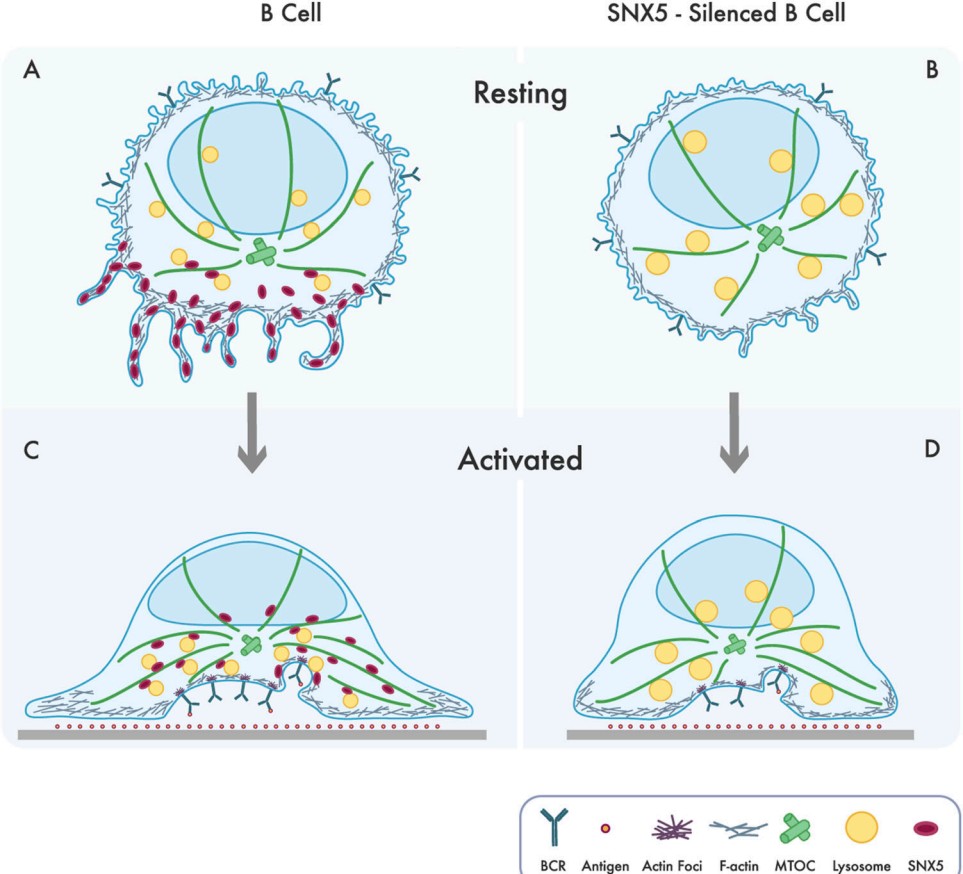

**Figure 6. Scheme depicting how endolysosomal dynamics and actin-dependent membrane spreading at the IS are regulated by SNX5.**
**(A)** Under resting conditions, SNX5 is mainly localized to ruffles. B-cell receptor engagement enhances its localization to endolysosomal compartments. **(B)** SNX5-silenced cells decrease membrane ruffle formation and exhibit alteration in lysosome homeostasis (changes in pH and increase in size). **(C)** B cells undergo spreading upon contact with immobilized antigens, where SNX5 is recruited to the synaptic membrane in close association with the B-cell receptor, endolysosomal compartments, and actin. **(D)** In the absence of SNX5, there is an impaired recruitment of lysosomes, reduced cell spreading, and actin foci at the IS, leading to decreased antigen uptake and presentation.

endolysosomal compartments to the IS in SNX5-silenced cells could be due to defects in their association with the microtubule network where SNX5 could act as adapter. Interestingly, endosomal membranes act as assembly sites for branched actin filaments that protrude outward from the surface of the endosome regulating endosomal sorting dynamics (34, 35). We showed that SNX5 also regulates the formation and/or stabilization of actin foci and contacts between actin and LAMP1$^+$ endolysosomal compartments, which could result in defects in the assembly of signaling components to lysosomes, also compromising antigen extraction at the IS. (2) Disruption of B-cell activation due to defects in the actin cytoskeleton.

Previous studies have shown that SNX5 is involved in maintaining phosphoinositides at the plasma membrane, which could aid in the formation of ruffles. In addition, its interaction with phosphoinositides regulates the endosomal sorting of receptors, such as EGFR (36, 37). In B cells activated with surface-tethered antigens, SNX5 was recruited to the IS in close association with BCR. However, the absence of SNX5 did not affect the internalization and turnover of BCR at the plasma membrane. Interestingly, during B-cell activation, the absence of SNX5 leads to reduced cellular spreading responses, which highlights the role of SNX5 in actin remodeling, as observed in other immune cells (11).

In parallel, the activation of the BCR leads to the accumulation of phosphatidylinositol phosphates at the synaptic membrane (38),

which interact with SNX5 through its BAR domain. Interestingly, SNX5 can also bind to DOCK family proteins that possess a DHR1 domain (39). The DOCK family of proteins act as adapters for signaling molecules that regulate the actin cytoskeleton, such as WASP and Arp2/3 (26, 40). Thus, we speculate that SNX5 could bind to phosphatidylinositols at the plasma membrane and mediate the anchoring of DOCK proteins through DHR1. This would allow the recruitment of WASP and Arp2/3 modulating actin polymerization, which could be enhanced by the accumulation of HS1 and Erk at the IS. How SNX5 orchestrates the recruitment and function of effectors that regulate actin remodeling at the IS remains to be addressed.

Overall, SNX5 plays a crucial role in B-cell activation by promoting actin remodeling at the plasma membrane during the search for antigens and maintaining LAMP1$^+$ endolysosomal homeostasis, required to facilitate the extraction of immobilized antigens at the immune synapse.

# Materials and Methods

### Cell culture

The mouse IgG$^+$ B-lymphoma cell line, IIA1.6, was used (41). The cells were cultured in CLICK medium (RPMI 1640 with GlutaMAX

supplemented with 1 mM sodium pyruvate, 100 U/ml penicillin, 100 $\mu$g/ml streptomycin, 0.1% $\beta$-mercaptoethanol, and 10% FBS) in a cell culture incubator (37°C/5% $CO_2$).

## Antibodies

The following primary antibodies were used for immunofluorescence: F(ab')$_2$ goat anti-mouse immunoglobulin G (IgG) (Jackson ImmunoResearch), rabbit anti-SNX5 (#180520, 1:200; Abcam), rat anti-$\alpha$-tubulin (#ab6160, 1:500; Abcam), rat anti-(LAMP1) CD107a (#553792, 1:200; BD Biosciences), rabbit anti-OVA (#C6534, 1:500; Sigma-Aldrich), chicken anti-GFP (#ab13970; Abcam), and rabbit anti-pHS1 and HS1 (#4557S and #8714S, 1:200; Cell Signaling).

The following secondary antibodies were used: Alexa Fluor 488–conjugated goat anti-rabbit (1:200; LifeTech), Alexa Fluor 647– and Cy3–conjugated F(ab)$_2$ donkey anti-rat and Cy3-conjugated F(ab)$_2$ donkey anti-rabbit (1:200; Jackson ImmunoResearch), rhodamine phalloidin (#R415, 1:200; Invitrogen), Alexa Fluor 488 F(ab')$_2$ donkey anti-chicken IgG (H+L) (#703-546-155, 1:300; Jackson ImmunoResearch), Hoechst (#33342, 1:1,000; Thermo Fisher Scientific), F(ab')$_2$ Alexa Fluor 546–conjugated goat anti-mouse IgG and Alexa Fluor 647–conjugated F(ab')$_2$ goat anti-mouse IgG (#a11018; Invitrogen), and Alexa Fluor 488–conjugated F(ab')$_2$ goat anti-mouse IgG (Jackson ImmunoResearch).

For Western blot, the following antibodies were used: rabbit anti-SNX5 (#180520, 1:1,000; Abcam), rat anti-$\alpha$-tubulin (#ab6160, 1:1,000; Abcam), and rabbit anti-EXOC7 (#ab95981, 1:1,000; Abcam); GAPDH (#G9545 1:1,000; Sigma-Aldrich); and Erk and pErk (#4695s and #9101s, respectively, 1:1,000). The following secondary antibody was used: HRP-conjugated donkey anti-rat, anti-rabbit, or anti-mouse (1:5,000; Jackson ImmunoResearch).

## Cell transfection

Amaxa Cell Line Nucleo-factor Kit R (L-013 program; Lonza) was used to electroporate $5 \times 10^6$ cells with siRNA against SNX5 (#s87540 [siSNX5-A] and #s87541 [SNX5-B]; Life Technologies). As a control, we used a scrambled siRNA (QIAGEN) at 10 nM. Cells were cultured in CLICK medium for 16 h in a cell culture incubator before functional analysis.

Plasmid expression was achieved by electroporating $5 \times 10^6$ cells with 2 $\mu$g of plasmid using Amaxa Cell Line Nucleo-factor Kit R (L-013 program; Lonza). SNX5-GFP was obtained by OriGene (NM_024225), and Rab5-YFP and GFP plasmid were kindly provided by Alfonso González (Universidad San Sebastián, Chile).

## B-cell activation using Ag-coated beads, Ag-coated slides, or soluble antigen

### Preparation of Ag-coated beads
$2 \times 10^7$ 3-$\mu$m latex NH$_2$ beads (PolyScience) were activated with 8% glutaraldehyde (Sigma-Aldrich) for 4 h at RT. Beads were washed with phosphate-buffered saline and incubated overnight at 4°C with 100 $\mu$g/ml of F(ab')$_2$ goat anti-mouse IgG (antigen, BCR ligand$^+$ beads) or 100 $\mu$g/ml of F(ab')$_2$ goat anti-mouse IgM (BCR ligand$^-$ beads). For antigen extraction assays, beads were coated with BCR

ligand$^+$ or BCR ligand$^-$ plus or not with 100 $\mu$g/ml of the *L. major* antigen Lack or OVA, as previously described by reference 6.

### Preparation of Ag-coated cover slides
Slides were incubated with 0.1 $\mu$g/$\mu$l of F(ab')$_2$ goat anti-mouse IgG (BCR ligand$^+$) overnight at 4°C in PBS as previously described by reference 6.

### Preparation of soluble antigen
Immunocomplexes were prepared as previously described by reference 42. For BCR cross-linking activation, cells were activated with 10 $\mu$g/ml of F(ab')$_2$ goat anti-mouse IgG premixed with 20 $\mu$g/ml of F(ab')$_2$ donkey anti-goat IgG incubated for 1 h at RT.

## B-cell activation and immunofluorescence

### Activation with soluble antigen
Cells were activated with soluble antigen in the form of an immunocomplex (F(ab')$_2$ goat anti-mouse IgG + F(ab')$_2$ donkey anti-goat IgG) for different time points. The reaction was stopped by adding ice-cold PBS/2% BSA. For time 0, the samples were kept on ice all the time. After activation, cells were washed with ice-cold PBS 1X and fixed in 3% PFA for 10 min on ice. Then, cells were blocked with 0.3 M of glycine blocking buffer, permeabilized with permeabilization buffer (PBS 1X/0.2% BSA/0.05% saponin) for 10 min, and incubated with primary antibodies in permeabilization buffer overnight. Secondary antibodies were incubated for 60 min as previously described by reference 6. Coverslips were mounted on slides using 10 $\mu$l Fluoromount-G (Electron Microscopy Sciences). For immunofluorescence of B cells transfected for Rab11-YFP and Rab5-YFP, the signal was enhanced using chicken anti-GFP as a primary antibody and Alexa Fluor 488 F(ab')$_2$ donkey anti-chicken IgG as a secondary antibody.

### Activation with immobilized antigens
Cells were activated with BCR ligand$^+$ beads at a 1:1 ratio, and plated on poly-L-lysine–coated cover slides or onto BCR ligand$^+$–coated glass coverslips for different time points in a cell culture incubator. Cells were washed and fixed in 3% PFA for 10 min at RT; for $\gamma$-tubulin staining, cells were fixed with methanol for 1 min on ice. Then, cells were incubated with primary/secondary antibodies in permeabilization buffer. Coverslips were mounted on slides using 10 $\mu$l Fluoromount-G.

### LysoSensor staining
Control and SNX5-silenced B cells were washed and incubated with LysoSensor Green DND-189 (#L7535, 1:500; Thermo Fisher Scientific) for 1 h in a cell culture incubator. LysoSensor was detected using live-cell epifluorescence microscopy.

## Antigen internalization by flow cytometry

B cells were activated with soluble antigen at 37°C for different time points. For time 0, samples were kept on ice. Then, cells were washed with ice-cold PBS/2% BSA, fixed with 3% PFA, washed, and analyzed. Cell surface BCR levels were assessed by flow cytometry

as mean fluorescence intensity (LSRFortessa X-20). Data were analyzed using FlowJo v10 (Tree Star).

## BCR surface and SNX5 levels by cytometry

Control and SNX5-silenced B cells were stained on ice for 10 min with F(ab)$_2$ goat $\alpha$-mouse IgG-488, washed with ice-cold PBS/2% BSA, and fixed with 3% PFA. Then, cells were permeabilized with permeabilization buffer (PBS/0.2% BSA/0.05% saponin) for 10 min, incubated with anti-SNX5 antibody for 2 h and secondary antibody Alexa Fluor 488–conjugated goat anti-rabbit for 1 h in permeabilization buffer, washed, and analyzed as mean fluorescence intensity for each condition.

## Ag extraction assays

OVA and BCR ligand$^+$ were coupled to NH$_2$ beads in equal concentrations (100 $\mu$g/ml each). Cells were incubated in a 1:1 ratio with antigen-coated beads plated on poly-L-lysine cover slides at 37°C for indicated times, fixed, and stained for OVA in permeabilization buffer overnight.

## Antigen presentation assay

B cells were incubated with Lack and BCR$^+$ ligand or BCR$^-$ ligand–coated beads; or preprocessed peptide for 4 h. Cells were then washed with PBS, fixed in ice-cold PBS/0.01% glutaraldehyde for 1 min, and quenched with PBS/100 mM glycine. B cells were incubated with a Lack-specific T-cell hybridoma at a 1:1 ratio for 4–24 h (5, 6). Supernatants were collected, and IL-2 cytokine production was measured using the BD OptEIA Mouse IL-2 ELISA kit, following the manufacturer's instructions (BD Biosciences).

## Dextran capture assay

Control and SNX5-silenced B cells were incubated with Texas Red–conjugated 10 kD dextran for 1 h, washed, and chased for 4 h at 37°C. After incubation, cells were washed, fixed, and processed for immunofluorescence.

## Live imaging for ruffle formation

To assess ruffle formation, live imaging of B cells was performed under steady-state conditions every 1 min for 10 min, followed by incubation with soluble antigen for 10 min. To visualize SNX5 in protrusions, SNX5-GFP– and GFP-expressing B cells were imaged using epifluorescence microscopy. To assess ruffle formation in Ctrl and SNX5-silenced B cells, DIC microscopy was used. To decrease noise from epifluorescence microscopy, ROF denoise Theta 25.0 was used from Fiji software (43).

## Synaptic membrane isolation

Synaptic membrane isolation followed the protocol detailed by references 19, 44, and 5 × 10$^6$ B cells were activated in 300 $\mu$l of CLICK medium with 2% FBS containing BCR ligand$^+$ Dynabeads (Invitrogen) in a 1:1 cell-to-bead ratio for specified durations. Activation was stopped by adding 500 $\mu$l of ice-cold PBS, and the cells were centrifuged at 600$g$ for 5 min at 4°C. The cells were resuspended in 500 $\mu$l of freeze–thaw buffer (20 mM Tris–HCl, 600 mM KCl, 20% glycerol, 1 mM Na$_3$VO$_4$, 5 mM NaF, and protease inhibitor cocktail, pH 7.4) and lysed through seven cycles of freezing and thawing at –80°C. The lysates were incubated with Benzonase (Merck) for 10 min at RT, followed by washing with 500 $\mu$l of freeze–thaw buffer using a magnet to precipitate the synaptic membrane–Dynabeads complex. Samples were resuspended in loading buffer and analyzed by Western blot.

## Western blot

Cells were lysed for 30 min (4°C) with 0,1% $\beta$-mercaptoethanol, 0.5% NP-40, 1 mM Hepes (pH 7,2), 0.5 mM MgCl$_2$, and 1 mM PMSF protease inhibitor cocktail (Roche). Proteins were resolved on 10% polyacrylamide gels and transferred onto a polyvinylidene fluoride membrane (PDVF, Trans-Blot Semi-Dry Transfer Cell; Bio-Rad). Membranes were fixed in methanol and blocked in 5% BSA-TBS + 0.05% Tween-20 for 1 h. Antibodies were diluted in 2% BSA-TBS + 0.05% Tween-20 and incubated overnight. Secondary antibodies were incubated for 1 h at RT. Western blots were developed with Westar ηC 2.0 or Westar Supernova substrate (Cyanagen), and chemiluminescence was detected using G:BOX iChemi (Syngene).

## Immunofluorescence image acquisition

### Epifluorescence microscopy
All z-stack images were obtained with 0.5-$\mu$m slices. Images were acquired in an epifluorescence microscope (Nikon Ti2 Eclipse) with an objective X60/1.25NA.

### Confocal microscopy
Images were acquired in a confocal microscope (Nikon Time Lapse) with a CMOS camera (ORCA-FLASH 4.0 V3; Hamamatsu C13440) and a Zeiss Airyscan microscope (LSM 880 Zeiss with Airyscan detection), with 63X/1.4 NA and with a z-stack of 0.25–0.5 µm operated with NIS-Elements version 4.6 and Zen Blue software, respectively.

### DIC microscopy
Images were acquired in a Nikon Ti2 Eclipse inverted microscope with a 100x/1.50 NA oil immersion lens and an iXON Ultra EMCCD camera at 37∘C.

### TEM
For TEM, B cells were fixed with 3% glutaraldehyde at RT. Images were acquired from ultrathin sections using a JEOL-1010 electron microscope (JEOL USA, Inc.) with a SC1000 ORIUS-CCD digital camera (Gatan Inc.) from Universität of Barcelona, Spain.

### Image analysis
Image processing and analysis were performed with Fiji software (43). 3D analysis was performed using the SCIAN-Soft software tools (https://github.com/scianlab), programmed on the IDL programming

language (ITT/Harris Geospatial). Image brightness and contrast were manually adjusted for visualization purposes but not for analysis (e.g., in the case of segmentations).

### Polarity analysis

MTOC, SNX5, and LAMP1 polarity indexes were determined as previously described (45). Briefly, we manually selected the location of the MTOC and delimited the cell and we also obtained the center of mass of the cell (Cellmc) and the bead (Beadmc). The position of the MTOC was projected on the vector defined by the Cellmc and Beadmc axis (Pmtoc). The MTOC polarity index was calculated by dividing the distance between Cellmc and Pmtoc and the distance between Cellmc and Beadmc. The index ranged from −1 (anti-polarized) to +1 (fully polarized). LAMP1 and SNX5 polarity indexes were calculated using the center of mass of the fluorescence in the same manner mentioned for MTOC.

### Lysosomes around bead

The accumulation of lysosomes at the IS was quantified by measuring the LAMP1 fluorescence intensity within a concentric circular area (3.5 $\mu m$) surrounding the bead. This value was then normalized to the total LAMP1 fluorescence intensity of the entire cell.

### Antigen extraction analysis of B cells activated with beads

The amount of OVA present on the beads was calculated by defining a fixed area around beads in contact with cells and measuring fluorescence on three-dimensional projections of a z-stack. The percentage of antigen extracted was estimated by the percentage of fluorescence intensity lost by the beads after 2 h, as previously described (19).

### Spreading area

We manually delimited the border of the cell at the z-slice corresponding to the synaptic plane, using the phalloidin label as a template.

### BCR recruitment in B cells activated on Ag coverslips

BCR recruitment at the IS was calculated by defining a ratio of fluorescence density between the central and the total area of the z-slide at the immune synapse. The central area was defined as 1/4 of the size of the total outline. The border of the cell was delimited using the phalloidin label as a template, as mentioned above. The values obtained can vary from negative to positive, indicating a peripheral or central distribution of BCR, respectively, as was described previously (45).

### Actin accumulation on beads

Accumulation of F-actin on the beads was calculated by defining a fixed area around beads in contact with cells and measuring fluorescence on three-dimensional projections of a z-stack.

### Circularity

DIC was used for ruffle quantification in both control and SNX5-silenced B cells. Circularity of cells was calculated using an extended version of the Fiji software plugin, with the formula: Circularity = $4\pi(\text{area}/(\text{perimeter})^2)$.

### Number of LAMP1$^+$ endolysosomal compartments at the IS

The number of lysosomes was quantified by Analyze Particles (size = 0.10–4.00 $\mu m^2$) present in the z-slice corresponding to the synaptic plane.

### LAMP1, SNX5, or actin volume/number

Confocal image stacks were first filtered in Fiji using the Trainable Weka Segmentation plugin (46), to enhance membrane signals. Next, 2D threshold filters were applied within SCIAN-Soft (47) to produce binary images of the LAMP1, SNX5, or actin fluorescence signal as regions of interest (ROIs).

### Actin intersection with LAMP1

LAMP1–actin ROIs were defined by a logical filter to obtain the intersection between the actin ROIs and LAMP1 ROI. In each case (LAMP, actin, and LAMP1–actin), adjacent ROIs along the z-axis were connected to define 3D ROIs, and their volume was quantified by voxel counting as was described by reference 47.

### LysoSensor and pHS1 intensity

Fluorescence of LysoSensor was quantified from SUM slices from z-projections, and the fluorescence of pHS1 was calculated considering the stack corresponding to the synaptic plane.

### Colocalization analysis of dextran with LAMP1

Pearson's colocalization analysis was performed considering each cell z-stack.

### Quantification of LAMP1$^+$ endolysosomal compartments containing antigens and SNX5

To quantify LAMP1$^+$ endolysosomal compartments containing antigens and SNX5, we performed 3-dimensional intersections computed with the following steps: (i) segmented images of LAMP1, soluble antigen, and SNX5 were obtained with Fiji software, a random forest classification model, trained within Fiji with the Weka plugin (46), and the segmented images were eroded in 1 pixel to refine boundaries; (ii) 3D voxel models for each marker were generated in SCIAN-Soft (47) by z-slice stacking of the segmented images; (iii) voxel-level intersections of the LAMP1 and antigen, LAMP1 and SNX5, and antigen and SNX5 aggregates were computed pairwise for the respective 3D models; (iv) finally, subsequent LAMP1+ compartment selections were performed from the intersection voxel computation: the volume of LAMP1+ lysosome compartments that contain antigen (LAMP1+Ag+) was selected and normalized by the total LAMP1+ volume (percentage of LAMP1+Ag+ relative to LAMP1+ compartments). A second selection was then performed to identify which volume of LAMP1+Ag+ compartments intersected with SNX5 (LAMP1+Ag+SNX5+), relative to the volume of LAMP1 intersected with SNX5 compartments (LAMP1+SNX5+).

## Statistical analysis

Data were collected from three independent experiments, except when specified otherwise. Statistical analysis was performed with GraphPad Prism 9 (GraphPad Software). Data are generally reported as the mean ± SEM and analyzed by a t test or one- or two-way ANOVA. The P-values were computed using different tests as

indicated in figure legends: * 0.01 < *P* < 0.05; **0.001 < *P* < 0.01; and ***P* < 0.001.

## Supplementary Information

## Acknowledgements

We thank Unidad de Microscopía Avanzada (UMA) and Unidad de Citometría from Pontificia Universidad Católica de Chile, especially Nicole Salgado and Fernanda Gárate for their support in image acquisition and Alex Cabrera and Rancés Blanco for their support with cytometry; Carlos Enrich and Carles Rentero for support in TEM from University of Barcelona; Unidad de Microscopía Electrónica de los Centros Científicos y Técnicos de la Universidad de Barcelona (CCiT/UB) del Campus Casanova, Facultad de Medicina y Ciencias de la Salud; Gabriela Cifuentes-Couchot for generating the graphic summary model; and Jonathan Lagos, Jorge Ibañez, Juan José Saez, Martina Alamo, and Felipe Del Valle for critical feedback and ideas. This project was supported by research grants from FONDECYT 3220832 to J Jara-Wilde, FONDECYT 1171024 to J Díaz-Muñoz, FONDECYT 1221128 to M-I Yuseff, Embo Global Investigator-GIN-4461 to M-I Yuseff, and ANID scholarship program 21220609 to F Cabrera-Reyes.

### Author Contributions

F Cabrera-Reyes: conceptualization, data curation, formal analysis, validation, investigation, visualization, methodology, project administration, writing—original draft, review, and editing, and performed most of the experiments, analysis, and wrote the manuscript.
T Contreras-Palacios: formal analysis, validation, methodology, and performed experiments and helped draft the manuscript.
R Ulloa: formal analysis, validation, methodology, and performed experiments and helped draft the manuscript.
J Jara-Wilde: software, formal analysis, methodology, and provided tools and feedback for 2D and 3D analysis.
M Caballero: formal analysis, methodology, and performed immunoblots and analysis.
C Quiroga: resources, validation, methodology, and provided tools for analysis and development of the project.
CG Feijoo: resources, validation, methodology, and provided tools for analysis and development of the project.
J Díaz-Muñoz: conceptualization, resources, data curation, software, formal analysis, supervision, funding acquisition, validation, investigation, visualization, methodology, project administration, and writing—original draft, review, and editing.
M-I Yuseff: conceptualization, resources, data curation, software, formal analysis, supervision, funding acquisition, validation, investigation, visualization, methodology, project administration, and writing—original draft, review, and editing.

### Conflict of Interest Statement

The authors declare that they have no conflict of interest.

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
