## [Reviewer comments · Life Science Alliance]

Life Science Alliance

SNX5 Promotes Antigen Presentation in B Cells by Dual Regulation of Actin and Lysosomal Dynamics

Fernanda Cabrera-Reyes, Teemly Contreras-Palacios, Romina Ulloa, Jorge Jara-Wilde, Mia Caballero, Clara Quiroga, Carmen G Feijoo, Jheimmy Díaz-Muñoz, and Maria-Isabel Yuseff

DOI: <https://doi.org/10.26508/lsa.202402917>

Corresponding author(s): Maria-Isabel Yuseff, Pontificia Universidad Católica de Chile and Jheimmy Díaz-Muñoz, Pontificia Universidad Católica de Chile

Review Timeline:

Submission Date:	2024-07-03
Editorial Decision:	2024-08-15
Revision Received:	2024-10-08
Editorial Decision:	2024-10-10
Revision Received:	2024-10-15
Accepted:	2024-10-16

Transaction Report:

August 15, 2024

Re: Life Science Alliance manuscript #LSA-2024-02917

Dr. Maria-Isabel Yuseff
Pontificia Universidad Católica de Chile
Department of Cellular and Molecular Biology,
Av. B. O'Higgins 340
Santiago 8320000
Chile

Dear Dr. Yuseff,

Thank you for submitting your manuscript entitled "SNX5 couples actin-dependent membrane remodeling to antigen extraction in B cells" to Life Science Alliance. The manuscript was assessed by expert reviewers, whose comments are appended to this letter. We invite you to submit a revised manuscript addressing the Reviewer comments.

Thank you for this interesting contribution to Life Science Alliance. We are looking forward to receiving your revised manuscript.

Sincerely,

B. MANUSCRIPT ORGANIZATION AND FORMATTING:

Reviewer #1 (Comments to the Authors (Required)):

Summary: This manuscript provides important new insights into the multiple roles of the sorting nexin SNX5 in B cells. The authors elegantly demonstrate that SNX5 links membrane dynamics, cytoskeletal organization, lysosome dynamics and polarization, and BCR-mediated antigen extraction. The role of SNX proteins had not been previously studied in B cells and this manuscript implicates SNX5 as a key integrator of B cell function. Prior to antigen encounter, SNX5 is localized near the plasma membrane and supports membrane ruffling, which may enhance the ability of B cells to scan their environment for antigens. SNX5 also maintains lysosome homeostasis in B cells. Upon BCR engagement with an anti-Ig antibody (a surrogate antigen), SNX5-dependent membrane ruffling is terminated, and SNX5 then plays key roles in building actin structures at the immune synapse, cell spreading, recruiting lysosomes to the sites of BCR clustering, and promoting lysosome-dependent extraction of antigens from surfaces. Several mechanistic connections are revealed, including the role of SNX5 in promoting the phosphorylation/activation of HS1, which supports Arp2/3 complex-mediated actin polymerization. These are new findings that will spur further research. The discussion section is particularly informative and forward-looking, and the summary figure/graphical abstract is very helpful. The data are of high quality and the 3D volumetric analysis of fluorescence microscopy images is impressive. In most cases, representative images are paired with quantitative analyses from an appropriate number of cells. The manuscript is well-written. However, some wording could be more precise and more information could be provided about how certain parameters are measured. Most of my comments are aimed at making the data easier for the reader to interpret and understand.

A. Points that could potentially be addressed by additional experiments:

1. Figure S4 and lines 286-288: "... upon activation with BCR ligand, SNX5-silenced B cells displayed lower levels of phosphorylated Erk (pErk) compared to control cells confirming that defects in BCR downstream signaling (Supplementary Fig S4A)". Although it looks like BCR-induced ERK activation is less sustained in the SNX5-silenced cells, these results seem preliminary. Quantification of results from multiple blots would strengthen this point. However, this is not the main focus of the manuscript and further analysis of the effects of SNX5 on BCR signaling could be pursued in subsequent work.

2. In Figure 2A, the positions of the MTOCs are not obvious. Pericentrin staining would provide more precise and obvious localization of the MTOC. Otherwise, the authors should add arrows to these images to identify the MTOC positions that they used for quantifying the polarity indexes.

B. Points that can be addressed in the text:

1. Line 54. It would be good to cite some recent review articles describing the B cell immune synapse.

2. Lines 107-108 and Figure 1B. The authors should clearly indicate that the soluble "antigen" they are using is a Fab'2 of an anti-IgG antibody. The recent structural studies by Reth and colleagues have suggested that anti-Ig antibodies do not cause the same activating structural changes in BCR complexes as occupancy of the antigen-binding sites. The authors should clearly state that the BCR ligand or "antigen" that is used throughout the manuscript is an anti-Ig antibody.

3. Lines 151-157, Figures 1F-1G, and lines 875-876 (legend for figure 1G).

a) I found the nomenclature and wording to be somewhat confusing. SNX5 and Ag are superscripted but LAMP1 is not. It would be much clearer if flow cytometry-style nomenclature were used, e.g., LAMP1⁺ SNX5⁺ Ag⁺ compartments, and if the rows in Figure 1G were indicated as:

$\%(\text{LAMP1}^+ \text{ Ag}^+ \text{ volume}) / [(\text{LAMP1}^+ \text{ Ag}^- \text{ volume}) + (\text{LAMP1}^+ \text{ Ag}^+ \text{ volume})]$

$\%(\text{LAMP1}^+ \text{ SNX5}^+ \text{ Ag}^+ \text{ volume}) / [(\text{LAMP1}^+ \text{ SNX5}^+ \text{ Ag}^- \text{ volume}) + (\text{LAMP1}^+ \text{ SNX5}^+ \text{ Ag}^+ \text{ volume})]$

b) On lines 151-157, the authors describe these ratios as the "percentage of lysosomes", which suggests that individual lysosomes were counted and characterized as being positive or negative for Ag. I'm not sure that this is what was done. Do the authors mean "the percentage of LAMP1⁺ organelle volume that co-localized or intersected with Ag staining"?

c) In this regard the term "intersect" could be confusing and should be distinguished from "co-localization", which implies overlap of fluorescence signals. "Intersect" could mean counting all of the Ag fluorescence in a vesicle/compartament that has only a

small overlap with LAMP1 staining or just contacts a LAMP1-stained compartment. This is particularly evident in Figure 5A, where the actin compartments are touching the LAMP1 compartments but there is no spatial overlap. Upon first usage, a clear explanation of how the term "intersect" is being used is essential.

d) Figure 1F would be easier for readers to interpret if the image of the LAMP1+ Ag+ compartments in the rightmost panel were a different red/blue merged color than the red color that is used for LAMP1 staining in the leftmost panels. Also, the authors imply that all of the LAMP1+ volume is also SNX5+. To support that, another panel should be added to show the 3D SNX5 staining.

e) In Figure 1G, it would be good to title the columns "early activation (10 min)" and "late activation (60 min)" to match figure 1F.

f) The legends to figure 1F and 1G should indicate that these are representative images from 1 cell. A statement about reproducibility, e.g., "similar results were obtained from X cells in X independent experiments" should be added. Although this is a complex low-throughput analysis, a supplemental figure showing similar results for 1-2 additional cells from an independent experiment would strengthen the conclusion, which is currently based on images from 1 cell. Similar statements should be added to other figures where representative images are shown without additional quantification.

4. In Figure 2A, it would be better to identify the stimuli as "anti-IgG-coated beads" and "control beads". The nature of the control beads needs to be defined. Were they coated with an irrelevant Fab'2?

5. Lines 201-203. Perhaps the following statement could be qualified/softened since only a small fraction of SNX5 co-localizes with the microtubule staining: "At the IS, SNX5 also localized closely to the microtubule network, suggesting that a pool of SNX5 regulates vesicle trafficking through the microtubule network (Supplementary Fig S3C)." In Figure 6 (which is very nice!), the authors show SNX5 moving vesicles along non-cortical microtubules that link the MTOC to the plasma membrane. This is reasonable speculation but none of the images in Figure 2A or Figure S3C show this. In these images, the microtubules are all cortical and there is some SNX5 embedded in this cortical microtubule layer.

6. Lines 231-236. LLOme and other lysosome-disrupting agents generally neutralize/increase lysosomal pH. Perhaps the authors meant to say that both LLOme and SNX5 silencing increased lysosomal pH, instead of decreasing it? Indeed, on lines 366-368 in the discussion, the authors state that loss of SNX5 rendered the lysosomes less acidic.

7. Figure 3G and lines 242-245. The authors should clarify the criteria for determining whether or not LAMP1 lysosomes had formed a ring around the bead. Was a specific extent of LAMP1 contact with the bead or distance from the bead required for concluding that a LAMP1 ring had formed?

8. Lines 246-251. In the discussion, the authors could speculate whether the decreased recruitment of lysosomes to the IS in SNX5-silenced B cells is due to impaired MTOC polarization or to decreased association of lysosomes with the microtubule network.

9. Figure 3I. The text and figure legend state that the number of lysosomes in the confocal slice closest to the coverslip was determined but the methods section (Lines 587-589) indicates that the MFI of LAMP1 fluorescence was measured. Was this MFI somehow converted to number of lysosomes, or was the number of discrete LAMP1+ puncta quantified? The label on the Y-axis of the graph should reflect what was actually measured.

Reviewer #2 (Comments to the Authors (Required)):

The MS from Cabrera-Reyes et al. characterizes the roles of SNX5 during B cell activation. The authors show that SNX5 is present in the protrusions on resting B cells and later on LAMP+ endosomes after antigen extraction. Silencing SNX5 reduces cell spreading and antigen extraction, concomitantly affecting polymerized actin. Overall, the findings are novel (previous studies have only addressed the role of SNX5 in macrophages, as the authors have correctly cited) and are of basic importance in the B cell activation process. However, there are several accounts where the manuscript can be improved:

1. While the authors have nicely characterized multiple effects of SNX5 silencing, their claim of SNX5 "coupling" antigen extraction and actin remodeling is not truly investigated in the paper. Given the lack of a molecular mechanism, the two roles of SNX5 can very well be independent. I would strongly suggest that the authors alter their title to accommodate this disparity. Perhaps "shared" will be a better word to use.
2. Some of the author's claims need to be rephrased. This includes their interpretation of the SNX function (different effects of SNX5 knockdown on cells emanating from one common role in B cell activation needs reconsideration), SNX distribution (e.g. "at the IS, SNX5 also localized closely to the microtubule network, suggesting that a pool of SNX5 regulates vesicle trafficking through the microtubule network" is a tall claim and needs reevaluation given the colocalization is not visibly strong in the images and is not quantified), and the pH-sensing mechanism of their probe, Line 225 onwards).
3. I would also strongly recommend that the authors quantify the phenotypes as much as possible. These include the distribution

of SNX5 at ultrastructural level as well as in live cells (Fig1 A, B); western blot in Fig 2D, and molecular distribution in Fig 2D and F. Similarly, whole cell F-actin quantification is also missing. In addition, most of the images need to be magnified to examine distribution meaningfully. One key phenotype that requires magnified images is the presence of protrusions. In the current images, the cell protrusions are not clear. As such the connection between the protrusions sustaining via SNX5 and their absence in the siSNX5 cells doesn't appear to be very strong from the current images.

4. There are several careless errors in the text as well as images (e.g. Fig 2F has inconsistency in legend text and the image; Fig 5 is incorrectly labeled, and half of the legend is missing from the text; There is no Fig 4J and 4K- The pH51 figures are 4H and 4I.). Similarly, there are some inconsistencies in the quantification techniques also (two different colocalization algorithms have been used in two different figures). At the very least, an explanation of the rationale for choosing these methods should be provided.

Reviewer #3 (Comments to the Authors (Required)):

The work by Cabrera-Reyes et al. identifies a role for the sorting nexin SNX5 in coupling actin-dependent B cell morphology and antigen extraction, two crucial processes for B cell activation. Using elegant fluorescence microscopy combined with quantitative image analysis, they characterise SNX5 subcellular localisation including its distribution in antigen trafficking compartments and the immune synapse. They find that in the absence of SNX5, B cells have impaired membrane ruffling, recruitment of endolysosomes to the immune synapse, actin-dependent cell spreading, and extraction of immobilized antigen. This study provides interesting insights into B cell biology, the data are of high quality and support the main claims, and the paper is well written. It is well suited for publication in LSA.

Comments:

1. For most of the quantifications, the authors pool the cells from independent experiments and (it appears) calculate P values based upon the number of cells, rather than the number of independent experiments. Can the authors please correct or clarify this. Ideally data points would also be color-coded by experiment to visually highlight day-to-day variability in the measurements.
2. Fig. 1G: Please include mean +/- SEM ranges in the table from the biological replicates.
3. Does SNX5 depletion impact F-actin levels, or just F-actin remodelling/distribution? In the discussion it is suggested that SNX5 might maintain phosphoinositides at the plasma membrane, which would be very interesting. Alterations in phosphoinositides could affect both F-actin phenotypes.
4. The authors suggest that SNX5-silenced B cells may have defective BCR signalling, using reduced BCR recruitment to the cell center and a phospho-Erk blot as evidence (Fig. S4A). However, just looking at the blot, it is not obvious that this is the case. Because BCR signalling is so important to actin-dependent cell morphology and antigen extraction, these data should be quantified before making conclusions.
5. The data in Fig. 5C,D are interesting. Do the authors observe that the OVA fluorescence lost from the bead is being transported into the cell? It is difficult to tell from the images. Have the authors attempted to measure OVA peptide presentation to complement these experiments?

Minor comments:

1. The sentence beginning "this complex, including vacuolar protein sorting proteins..." on line 80 is unclear and should be rewritten.
2. Please identify the lymphoma line used in line 101.
3. Please define the soluble antigen used in line 107, and clarify that it is an anti-Ig surrogate antigen.
4. Line 123: "labelled filamentous actin (F-actin)"
5. The description of cell labelling on lines 138-139 is unclear; please rewrite this sentence.
6. Line 180 and 182: immunoblotting appears in Fig. 2D only (not Fig. 2E)
7. Line 189: Change Fig. 2F to Fig. 2E
8. Lines 299 and 300: Fig. 4J and 4K should be Fig. 4H and 4I.
9. Line 366-367: ...which were less acidic "and" were unable to be recruited efficiently...
10. Line 543: "objective" missing
11. Line 575: Please clarify how the central cell area was defined in the analysis.
12. Line 587: It is unclear how lysosome numbers were quantified. The methods state MFI, but the authors also include data showing that lysosome volume changes in SNX5-silenced cells. Please provide a more thorough description of the analysis methods used.
13. Please change red/green color schemes for images to magenta/green to make them accessible to readers with color blindness.

We thank the reviewers for their constructive feedback, which has greatly improved the quality of our manuscript. All the major points you raised have been addressed experimentally and included in the new version of our manuscript, as discussed below:

1-We show that B lymphocytes silenced for SNX5 display impaired antigen presentation, which supports our data showing that under these conditions, B cells extract less antigen from beads.

2-Quantitative imaging analysis of centrosome polarization in control and SNX5-silenced B cells, which shows that this process is not impaired.

3- Quantitative imaging analysis the recruitment of lysosome to the immune synapse of control and SNX5-silenced B cells, confirming that their accumulation is deficient.

4- Quantification of the recruitment of SNX5 to the synaptic membrane of activated B cells, shown by western blot.

Overall, we revised, added references and methodological information to our manuscript, according to the requests made by referees. Additionally, we improved the visualization of graphs adding arrows and making them accessible to readers with color blindness. We therefore hope that you will find our revised manuscript now suitable for publication in Life Science Alliance.

Reviewer 1

1. Figure S4 and lines 286-288: "... upon activation with BCR ligand, SNX5-silenced B cells displayed lower levels of phosphorylated Erk (pErk) compared to control cells confirming that defects in BCR downstream signaling (Supplementary Fig S4A)". Although it looks like BCR-induced ERK activation is less sustained in the SNX5-silenced cells, these results seem preliminary. Quantification of results from multiple blots would strengthen this point. However, this is not the main focus of the manuscript and further analysis of the effects of SNX5 on BCR signaling could be pursued in subsequent work.

Response: We have considered the reviewer's comments regarding ERK signaling in SNX5-deficient cells and have repeated this experiment. The supplementary figure now shows a quantification of 2 experiments. Also, in the results section (lines 305-308) we state more precisely that this result suggests that sustained BCR signaling could be regulated by SNX5.

2. In Figure 2A, the positions of the MTOCs are not obvious. Pericentrin staining would provide more precise and obvious localization of the MTOC. Otherwise, the authors should add arrows to these images to identify the MTOC positions that they used for quantifying the polarity indexes.

Response: We added arrows to the images (Figure 2A) to indicate the position of the MTOC in each cell.

1. Line 54. It would be good to cite some recent review articles describing the B cell immune synapse.
2. Lines 107-108 and Figure 1B. The authors should clearly indicate that the soluble "antigen" they are using is a Fab² of an anti-IgG antibody. The recent structural studies by Reth and colleagues have suggested that anti-Ig antibodies do not cause the same activating structural changes in BCR complexes as occupancy of the antigen-binding sites. The authors should clearly state that the BCR ligand or "antigen" that is used throughout the manuscript is an anti-Ig antibody.

Response: We appreciate the reviewer's comments and have added new references (line 57) and clarified the soluble antigen we used in results and methods section. We performed BCR cross-linking activation, where cells were activated with 10 µg/ml of F(ab')₂ goat anti-mouse IgG premixed with 20 µg/ml of F(ab')₂ donkey anti-goat IgG, mentioned in methodology section.

3. Lines 151-157, Figures 1F-1G, and lines 875-876 (legend for figure 1G).
 - a) I found the nomenclature and wording to be somewhat confusing. SNX5 and Ag are superscripted but LAMP1 is not. It would be much clearer if flow cytometry-style nomenclature were used, e.g., LAMP1⁺ SNX5⁺ Ag⁺ compartments, and if the rows in Figure 1G were indicated as:
 $\frac{\%(\text{LAMP1}^+ \text{ Ag}^+ \text{ volume})}{[(\text{LAMP1}^+ \text{ Ag}^- \text{ volume}) + (\text{LAMP1}^+ \text{ Ag}^+ \text{ volume})]}$
 $\frac{\%(\text{LAMP1}^+ \text{ SNX5}^+ \text{ Ag}^+ \text{ volume})}{[(\text{LAMP1}^+ \text{ SNX5}^+ \text{ Ag}^- \text{ volume}) + (\text{LAMP1}^+ \text{ SNX5}^+ \text{ Ag}^+ \text{ volume})]}$
 - b) On lines 151-157, the authors describe these ratios as the "percentage of

lysosomes", which suggests that individual lysosomes were counted and characterized as being positive or negative for Ag. I'm not sure that this is what was done. Do the authors mean "the percentage of LAMP1+ organelle volume that co-localized or intersected with Ag staining"?

c) In this regard the term "intersect" could be confusing and should be distinguished from "co-localization", which implies overlap of fluorescence signals. "Intersect" could mean counting all of the Ag fluorescence in a vesicle/compartment that has only a small overlap with LAMP1 staining or just contacts a LAMP1-stained compartment. This is particularly evident in Figure 5A, where the actin compartments are touching the LAMP1 compartments but there is no spatial overlap. Upon first usage, a clear explanation of how the term "intersect" is being used is essential.

Response to point 3. a), b) and c): We thank the reviewer for his/her suggestions. For figure 3D, we have modified the figures, legends, and explained the methodology as proposed. In the results section, we added more details on the analyses and explained how we determined the intersection to select lysosomes (LAMP+) with or without Ag and SNX5 (lines 156-170). Additionally, we used cytometry-style nomenclature.

d) Figure 1F would be easier for readers to interpret if the image of the LAMP1+ Ag+ compartments in the rightmost panel were a different red/blue merged color than the red color that is used for LAMP1 staining in the leftmost panels. Also, the authors imply that all of the LAMP1+ volume is also SNX5+. To support that, another panel should be added to show the 3D SNX5 staining.

Response: We have added the panel showing 3D SNX5 staining.

e) In Figure 1G, it would be good to title the columns "early activation (10 min)" and "late activation (60 min)" to match figure 1F.

Response: We have included these titles to the columns.

f) The legends to figure 1F and 1G should indicate that these are representative images from 1 cell. A statement about reproducibility, e.g., "similar results were obtained from X cells in X independent experiments" should be added. Although this is a complex low-throughput analysis, a supplemental figure showing similar results for 1-2 additional cells from an independent experiment would strengthen the conclusion, which is currently based on images from 1 cell. Similar statements should be added to other figures where representative images are shown without additional quantification.

Response: We agree with the reviewer and have included these statements in figure legends.

4. In Figure 2A, it would be better to identify the stimuli as "anti-IgG-coated beads" and "control beads". The nature of the control beads needs to be defined. Were they coated with an irrelevant Fab'2?

Response: We clarified this in the results and methods sections. BCR ligand- beads containing F(ab')₂ goat anti-mouse IgM were used as control beads.

5. Lines 201-203. Perhaps the following statement could be qualified/softened since only a small fraction of SNX5 co-localizes with the microtubule staining: "At the IS, SNX5 also localized closely to the microtubule network, suggesting that a pool of SNX5 regulates vesicle trafficking through the microtubule network (Supplementary Fig S3C)." In Figure 6 (which is very nice!), the authors show SNX5 moving vesicles along non-cortical microtubules that link the MTOC to the plasma membrane. This is reasonable speculation but none of the images in Figure 2A or Figure S3C show this. In these images, the microtubules are all cortical and there is some SNX5 embedded in this cortical microtubule layer.

Response: We fully agree with the reviewer's suggestion and have clarified this paragraph. We explain that a pool of SNX5 vesicles is localized in close association with microtubules at the IS and under resting conditions (Supplementary figure 4S). We added a complementary figure showing that some SNX5 signals follow both the non-cortical and cortical microtubule networks in non-activated cells. In addition, we included an image of a representative primary B lymphocyte displaying SNX5 puncta aligned with a microtubule (supplementary figure 4).

6. Lines 231-236. LLOme and other lysosome-disrupting agents generally neutralize/increase lysosomal pH. Perhaps the authors meant to say that both LLOme and SNX5 silencing increased lysosomal pH, instead of decreasing it? Indeed, on lines 366-368 in the discussion, the authors state that loss of SNX5 rendered the lysosomes less acidic.

Response: We thank the reviewer for pointing out this mistake. Indeed, silencing of LLOme and SNX5 increased lysosomal pH. This has been corrected in the manuscript.

7. Figure 3G and lines 242-245. The authors should clarify the criteria for determining whether or not LAMP1 lysosomes had formed a ring around the bead. Was a specific extent of LAMP1 contact with the bead or distance from the bead required for concluding that a LAMP1 ring had formed?

Response: We added the lysosome polarity index plot (figure 3G), reinforcing our results showing that fewer lysosomes reach the antigen-coated bead in SNX5-silenced cells. In the materials and methods section, we added a description of the quantification of lysosomes around the bead.

8. Lines 246-251. In the discussion, the authors could speculate whether the decreased recruitment of lysosomes to the IS in SNX5-silenced B cells is due to impaired MTOC polarization or to decreased association of lysosomes with the microtubule network.

Response: We quantified the repositioning of the centrosome (Figure 3I) towards the IS and our results show this was not impaired. This information is now included in the results section (lines 261-263)

9. Figure 3I. The text and figure legend state that the number of lysosomes in the confocal slice closest to the coverslip was determined but the methods section (Lines

587-589) indicates that the MFI of LAMP1 fluorescence was measured. Was this MFI somehow converted to number of lysosomes, or was the number of discrete LAMP1+ puncta quantified? The label on the Y-axis of the graph should reflect what was actually measured.

Response: This is now figure 3K. The number of lysosomes was quantified as particles (size=0.10-4.00 μm^2) present in z-slice corresponding to the synaptic plane using Fiji. This information has been corrected in the methods section (lines 636-638).

Reviewer 2

The MS from Cabrera-Reyes et al. characterizes the roles of SNX5 during B cell activation. The authors show that SNX5 is present in the protrusions on resting B cells and later on LAMP+ endosomes after antigen extraction. Silencing SNX5 reduces cell spreading and antigen extraction, concomitantly affecting polymerized actin. Overall, the findings are novel (previous studies have only addressed the role of SNX5 in macrophages, as the authors have correctly cited) and are of basic importance in the B cell activation process. However, there are several accounts where the manuscript can be improved:

1. While the authors have nicely characterized multiple effects of SNX5 silencing, their claim of SNX5 "coupling" antigen extraction and actin remodeling is not truly investigated in the paper. Given the lack of a molecular mechanism, the two roles of SNX5 can very well be independent. I would strongly suggest that the authors alter their title to accommodate this disparity. Perhaps "shared" will be a better word to use.

Response: We agree with the reviewer and changed the titled accordingly to **"SNX5 Promotes Antigen Presentation in B Cells by Dual Regulation of Actin and Lysosomal Dynamics"**.

2. Some of the author's claims need to be rephrased. This includes their interpretation of the SNX function (different effects of SNX5 knockdown on cells emanating from one common role in B cell activation needs reconsideration), SNX distribution (e.g. "at the IS, SNX5 also localized closely to the microtubule network, suggesting that a pool of SNX5 regulates vesicle trafficking through the microtubule network" is a tall claim and needs reevaluation given the colocalization is not visibly strong in the images and is not quantified), and the pH-sensing mechanism of their probe, Line 225 onwards).

Response: We agree with the reviewer and have clarified this paragraph in our manuscript. We explain that there are SNX5 signals that show a microtubule-like localization in the IS and also under resting conditions. To reinforce this, we added a complementary figure (Figure 4S) showing that some SNX5 signals follow the non-cortical and cortical microtubule network in non-activated cells. In addition, we included an image of a representative primary B lymphocyte displaying SNX5 puncta aligned with a microtubule.

3. I would also strongly recommend that the authors quantify the phenotypes as much as possible. These include the distribution of SNX5 at ultrastructural level as well as in live cells (Fig1 A, B); western blot in Fig 2D, and molecular distribution in Fig 2D and F. Similarly, whole cell F-actin quantification is also missing. In addition, most of the images need to be magnified to examine distribution meaningfully. One key phenotype that requires magnified images is the presence of protrusions. In the current images, the cell protrusions are not clear. As such the connection between the protrusions sustaining via SNX5 and their absence in the siSNX5 cells doesn't appear to be very strong from the current images.

Response: We appreciate the reviewers' comments and have included the following information: We improved figure 1B showing cells with SNX5+ projections more clearly. It is difficult to quantify SNX5 at the ultrastructural level (EM) and we would have to do this with immunogold labeling. Instead, we labeled SNX5 using immunofluorescence and clearly observe its enrichment in protrusions, see magnification (Figure 1C). Considering that these disappear upon antigen stimulation it is difficult to quantify this phenotype. We provide quantitative results of SNX5+ and lamp1 compartments using immunofluorescence, which we believe provide sufficient data on the dynamic localization of SNX5.

4. There are several careless errors in the text as well as images (e.g. Fig 2F has inconsistency in legend text and the image; Fig 5 is incorrectly labeled, and half of the legend is missing from the text; There is no Fig 4J and 4K- The pHS1 figures are 4H and 4I.). Similarly, there are some inconsistencies in the quantification techniques also (two different colocalization algorithms have been used in two different figures). At the very least, an explanation of the rationale for choosing these methods should be provided.

Response: We thank the reviewer for pointing out these errors and have now corrected them in the text in the corresponding sections (highlighted in the manuscript). To clarify, we used Manders analysis when we evaluated cells transfected with Rab5 YFP, because Rab5 expression varies in each cell. Otherwise, we used Pearson analysis.

Reviewer 3

The work by Cabrera-Reyes et al. identifies a role for the sorting nexin SNX5 in coupling actin-dependent B cell morphology and antigen extraction, two crucial processes for B cell activation. Using elegant fluorescence microscopy combined with quantitative image analysis, they characterise SNX5 subcellular localisation including its distribution in antigen trafficking compartments and the immune synapse. They find that in the absence of SNX5, B cells have impaired membrane ruffling, recruitment of endolyosomes to the immune synapse, actin-dependent cell spreading, and extraction of immobilized antigen. This study provides interesting insights into B cell biology, the data are of high quality and support the main claims, and the paper is well written.

Comments:

1. For most of the quantifications, the authors pool the cells from independent experiments and (it appears) calculate P values based upon the number of cells, rather than the number of independent experiments. Can the authors please correct or clarify this. Ideally data

points would also be color-coded by experiment to visually highlight day-to-day variability in the measurements.

Response: We corrected our graphs and analyzed the data as suggested.

2. Fig. 1G: Please include mean +/- SEM ranges in the table from the biological replicates.

Response: This has now been included in the revised version of the manuscript.

3. Does SNX5 depletion impact F-actin levels, or just F-actin remodelling/distribution? In the discussion it is suggested that SNX5 might maintain phosphoinositides at the plasma membrane, which would be very interesting. Alterations in phosphoinositides could affect both F-actin phenotypes.

Response: Silencing SNX5 in B cells did not show changes in actin levels (FigS1A) and mainly affected actin remodeling.

4. The authors suggest that SNX5-silenced B cells may have defective BCR signalling, using reduced BCR recruitment to the cell center and a phospho-Erk blot as evidence (Fig. S4A). However, just looking at the blot, it is not obvious that this is the case. Because BCR signalling is so important to actin-dependent cell morphology and antigen extraction, these data should be quantified before making conclusions.

Response: We repeated the experiment regarding ERK signaling in control and SNX5-deficient cells, obtaining consistent results. Considering that this is not the main scope of the manuscript we show this figure (quantification of both blots) as supplementary data. Also, in the results section (lines 304-307) we state more precisely that this result suggests that sustained BCR signaling could be regulated by SNX5.

5. The data in Fig. 5C,D are interesting. Do the authors observe that the OVA fluorescence lost from the bead is being transported into the cell? It is difficult to tell from the images. Have the authors attempted to measure OVA peptide presentation to complement these experiments?

Response: The OVA signal decreases over the activation time due to the progressive degradation of antigens facilitated by lysosome secretion. We do not possess a T cell line specific for OVA peptide presentation on MHCII.

To evaluate antigen presentation, we incubated both control and SNX5-silenced cells with beads coated with a specific BCR ligand, as well as the Lack antigen from *Leishmania major*. To assess the ability to present MHC-II-peptide complexes derived from bead-associated Lack, we measured IL-2 secretion from activated T lymphocytes by a specific T cell hybridoma. As anticipated, the SNX5-silenced B cells showed a significant reduction in their capacity to present the bead-associated Lack antigen to T lymphocytes.

Minor comments:

1. The sentence beginning "this complex, including vacuolar protein sorting proteins..." on line 80 is unclear and should be rewritten.

Response: This was corrected

2. Please identify the lymphoma line used in line 101.

Response: This was included

3. Please define the soluble antigen used in line 107, and clarify that it is an anti-Ig surrogate antigen.

Response: We appreciate the reviewer's comments and have added new references and clarified the soluble antigen we used in the methods section. We performed BCR cross-linking activation, where cells were activated with 10 µg/ml of F(ab')₂ goat anti-mouse IgG premixed with 20 µg/ml of F(ab')₂ donkey anti-goat IgG, mentioned in methodology section.

4. Line 123: "labelled filamentous actin (F-actin)".

Response: This was corrected.

5. The description of cell labelling on lines 138-139 is unclear; please rewrite this sentence.

Response: This was corrected

6. Line 180 and 182: immunoblotting appears in Fig. 2D only (not Fig. 2E).

Response: This was corrected

7. Line 189: Change Fig. 2F to Fig. 2E.

Response: This was corrected

8. Lines 299 and 300: Fig. 4J and 4K should be Fig. 4H and 4I.

Response: This was corrected

9. Line 366-367: ...which were less acidic "and" were unable to be recruited efficiently.

10. Line 543: "objective" missing.

11. Line 575: Please clarify how the central cell area was defined in the analysis. 12. Line 587: It is unclear how lysosome numbers were quantified. The methods state MFI, but the authors also include data showing that lysosome volume changes in SNX5-silenced cells. Please provide a more thorough description of the analysis methods used.

Response: We thank the reviewer for pointing out these mistakes and have now corrected them in the text. We also include information on how lysosome numbers were quantified using Fiji and included the details in the methods section.

13. Please change red/green color schemes for images to magenta/green to make them accessible to readers with color blindness.

Response: We thank the reviewer for this comment and have changed the colors of images to make them accessible to readers with color blindness.

October 10, 2024

RE: Life Science Alliance Manuscript #LSA-2024-02917R

Dr. Maria-Isabel Yuseff
Pontificia Universidad Católica de Chile
Department of Cellular and Molecular Biology,
Av. B. O'Higgins 340
Santiago 8320000
Chile

Dear Dr. Yuseff,

Thank you for submitting your revised manuscript entitled "SNX5 Promotes Antigen Presentation in B Cells by Dual Regulation of Actin and Lysosomal Dynamics". We would be happy to publish your paper in Life Science Alliance pending final revisions necessary to meet our formatting guidelines.

- please be sure that the authorship listing and order is correct
- please upload your manuscript text as an editable doc file
- please upload both your main and supplementary figures as single files
- please add the ORCID ID for the secondary corresponding author-they should have received instructions on how to do so
- please add the Twitter handle of your host institute/organization as well as your own or/and one of the authors in our system
- please use the [10 author names, et al.] format in your references (i.e. limit the author names to the first 10)
- you may want to consider uploading Figure 6 as a Graphical Abstract rather than as a figure, but this is up to you

A. FINAL FILES:

B. MANUSCRIPT ORGANIZATION AND FORMATTING:

Sincerely,

October 16, 2024

RE: Life Science Alliance Manuscript #LSA-2024-02917RR

Dr. Maria-Isabel Yuseff
Pontificia Universidad Católica de Chile
Department of Cellular and Molecular Biology,
Av. B. O'Higgins 340
Santiago 8320000
Chile

Dear Dr. Yuseff,

Thank you for submitting your Research Article entitled "SNX5 Promotes Antigen Presentation in B Cells by Dual Regulation of Actin and Lysosomal Dynamics". It is a pleasure to let you know that your manuscript is now accepted for publication in Life Science Alliance. Congratulations on this interesting work.

DISTRIBUTION OF MATERIALS:

Again, congratulations on a very nice paper. I hope you found the review process to be constructive and are pleased with how the manuscript was handled editorially. We look forward to future exciting submissions from your lab.

Sincerely,
